



# Downward and upward revisions of Chinese emissions of black carbon and CO in bottom-up inventories are still required: an integrated analysis of WRF/CMAQ model and EMeRGe observations in East Asia in spring 2018

Phuc Thi Minh Ha[1], Yugo Kanaya[1], Kazuyo Yamaji[2], Syuichi Itahashi[3], Satoru Chatani[4], Takashi Sekiya[1], Maria Dolores Andrés Hernández[5], John Philip Burrows[5], Hans Schlager[6], Michael Lichtenstern[6], Mira Poehlker[7], Bruna Holanda[7]

[1]Research Institute for Global Change, JAMSTEC, Yokohama 236-0001, Japan,
[2]Graduate School of Maritime Sciences, Kobe University, Kobe 658-0022, Japan,
[3]Research Institute for Applied Mechanics Department of Geoenvironmental Dynamics, Kyushu University, Fukuoka 816-8580, Japan,
[4]National Institute for Environmental Studies, Tsukuba, Ibaraki 305-8506, Japan,
[5]Institut für Umweltphysik, Universität Bremen, Otto-Hahn-Allee 1, Bremen 28359, Germany,
[6]DLR Oberpfaffenhofen, Institut für Physik der Atmosphäre, 82234 Wessling, Germany,
[7]Max Planck Institute for Chemistry, Multiphase Chemistry Department, Hahn-Meitner-Weg 1, 55128 Mainz, Germany

*Correspondence to*: Phuc T. M. Ha (phucha@jamstec.go.jp)

**Abstract.** Accurate estimates of short-lived climate forcers (SLCFs) emissions are required to allow efficient strategies that mitigate climate change to be developed. However, there remain large uncertainties about emissions of SLCFs from Asia. We identified and improved the constraints of combustion-related emissions of black carbon (BC) and CO using the WRF/CMAQ model (v5.0.2) and the EMeRGe airborne observation data for East Asia in spring 2018. We performed case studies of air masses containing emissions from fires near Thailand and emissions from urban areas in the Philippines and China. Chinese emissions were analysed in depth. Unlike for observations at ground-based stations, the observations from aircraft used here would not have been strongly influenced by local emissions and near-surface processes. We confirmed that the GFEDv4.1s inventory provided accurate data for emissions from fires near Thailand. However, anthropogenic BC and CO emissions from the Philippines (REASv2.1) were negatively biased. Marked positive and negative differences were found for BC (+1.62 µg m$^{-3}$) and CO (−400 ppbv) from the HTAPv2.2z emission inventory for Chinese air masses, consistent with the results of previous ground-based studies. The Chinese BC/CO emission ratio, 3.5±0.1 ng m$^{-3}$ ppb$^{-1}$, calculated using data from airborne observations in the altitude range ~0.3–1 km also agreed with the ground-based results. Linearly scaling BC emissions using an observation/model ratio (E(BC) = 0.48±0.13) gave our best estimate of 0.65±0.25 (Tg BC) yr$^{-1}$. The calculated BC/CO and CO/CO$_2$ ratios led us to estimate that emissions from China are 166±65 (Tg CO) yr$^{-1}$ and 12.4±4.8 (Pg CO$_2$) yr$^{-1}$. The results suggested that downward and upward revisions of Chinese emissions of BC (−50%) and CO (+20%), respectively, are required in HTAPv2.2z emission inventory.



## 1 Introduction

Short-lived climate forcers (SLCFs) are gases or particles that, like long-lived greenhouse gases, alter the energy balance of the Earth. Developing adequate strategies for mitigating human-induced climate change requires possible ways of decreasing SLCF concentrations to be considered (Myhre et al., 2013; Szopa et al., 2021; IPCC AR6 WG1 Ch6). Carbon monoxide (CO) can remain in the atmosphere for one to four months and can cause indirect warming as a SLCF by removing hydroxyl radicals (OH), which is the primary oxidant of $CH_4$. CO participates in the catalysed formation of $O_3$ in the presence of sufficient $NO_x$ (NO and $NO_2$).

Black carbon (BC) is mostly emitted through incomplete combustion of carbon-based fuels in boilers, cooking stoves, and heating stoves that use raw coal. BC causes radiative heating of the atmosphere by absorbing sunlight and climate forcing by changing the albedo of snow and ice. Distributed as nanometre-sized particles, BC strongly negatively affects human health in megacities (Shindell et al., 2012; WHO 2021 new AQ guideline).

A better understanding of SLCFs is needed to allow strategies that decrease air pollution, limit man-made climate change, and promote sustainable development (Szopa et al., 2021). This is challenging because there are various SLCFs and the physical and chemical processes by which SLCFs modify the climate are complex.

Incomplete combustion during most anthropogenic activities leads to BC and CO emissions. BC and CO emissions in Asia contribute >50% of total BC and CO emissions worldwide (Hoesly et al., 2018; Szopa et al., 2021). It is therefore important to have accurate data for BC and CO emissions. Currently, both our understanding of BC and CO emissions and the accuracy of estimated BC and CO emissions need to be improved (Kurokawa et al., 2013). Our understanding of the responses of SLCF emissions to the establishment of techniques that decrease emissions in the last two decades in fast-growing Asian economies is insufficient (Chen and Chen, 2019; Kanaya et al., 2020; Ikeda et al., 2023; Zhang et al., 2022). Biomass-burning habits in Southeast Asia has become the main contributor of carbon emissions from forest fires in spring (Reid et al., 2013; Heald et al., 2003; Palmer et al., 2006; Johnston et al., 2012). Forest fires can occur because of lightning strikes or human activities.

Investigations and assessments of combustion-related BC and CO emissions in East Asia benefit from airborne measurements made in the Effect of Megacities on the Transport and Transformation of Pollutants at Regional and Global Scales (EMeRGe) campaign in March and April 2018. High altitude and long range (HALO) aircraft can capture data for a pollution plume with a typical travel time of 3–5 d from the source, so careful selection of air masses without substantial effects of precipitation during transport could provide information about emissions of the observed species including $CO_2$. Such information about emissions from urban areas in Asia can be estimated and compared with emission data acquired in previous studies (Suntharalingam et al., 2004; Tohjima et al., 2010; Wada et al., 2011). These studies of outflows from Asia have shown the importance of deep convection (Folkins et al., 1997), the cold fronts tied to middle latitude cyclones (Liu et al., 2003; Sawa et al., 2007), and orography (Liu et al., 2003) to spatial transport of pollutants.

Pollution outflows observed during the EMeRGe-Asia campaign have been partially used to evaluate the performance of the Chemical Atmospheric General Circulation Model for the Study of Atmospheric Environment and Radiative Forcing



(CHASER) v4.0 including multiphase processes involving reactive nitrogen species, and large negative biases for daytime HONO concentrations in the East Asian coastal region were found (Ha et al., 2023). Different features for Asian and European outflows recorded during EMeRGe have been found, indicating that BC and CO outflows caused by the Manila plume in the Philippines were underestimated and the BC/CO ratio for overall Asia (4.1 ng m$^{-3}$ ppb$^{-1}$) was overestimated in multi-model

simulations (Deroubaix et al., 2024a, b). However, detailed region-specific emissions have not been explored.

Bottom-up emission inventories for Asia, and particularly China, have improved markedly in recent years, e.g., the uncertainty range for BC emissions is narrowed to ±208% (Kurokawa et al., 2013; Lei et al., 2011; Li et al., 2017; Lu et al., 2011; Zhao et al., 2011, 2013; Zhang et al., 2009; Zheng et al., 2018). However, verification using independent observations is still required to constrain the uncertainty ranges. Various bottom-up inventories have been compiled, and Chinese CO$_2$ emission rates have

had uncertainties of −10% to +9% for all emission sectors in 2010 (Zhao et al., 2013), 31% for the Regional Emission Inventory for Asia (REAS) v2.1 in 2010 (Kurokawa et al., 2013; Li et al., 2017b), and 19% for the REAS v3.2 in 2015 (Kurokawa and Ohara., 2020). In contrast, Chinese emissions of CO have an uncertainty range of ±86% (Kurokawa et al., 2013; Li et al., 2017a, b; Zhang et al., 2009; Zhao et al., 2013; Zheng et al., 2018). Emissions of primary particles such as PM$_{2.5}$ and BC are subject to more uncertain emission factors for the residential sector and for fuel combustion in stoves in residences in China

than emissions of gaseous species (Bond et al., 2002; Zhao et al., 2013; Zhi et al., 2008). Bottom-up emission estimates have been found to be biased relative to observation-based (or top-down) estimates, particularly when local contexts have not been fully taken into account in a timely manner when making bottom-up emission estimates (Choi et al., 2020; Kanaya et al., 2020; Suntharalingam et al., 2004; Zhao et al., 2013). Emission inventories therefore need to be tested using independent observational data.

The overall aim of this study was to improve our understanding of BC, CO, and CO$_2$ emissions in Asia to improve Asian emission inventories. We focused on specific types of emissions in four regions: fire emissions (including of biomass burning) near the Gulf of Thailand (THL) and selected polluted urban areas in the Philippines (PHL), China (CHN), and Japan (JPN). The primary objective was to estimate BC, CO, and CO$_2$ emissions in CHN using a combined model–observation approach using the Weather Research and Forecasting / Community Multiscale Air Quality (WRF/CMAQ) version 5.0.2 modelling

system and EMeRGe airborne observations for East Asia. The performance of the WRF/CMAQ system at ground level, where near-ground processes could affect the results, was evaluated using data from previous studies (Choi et al., 2020; Kanaya et al., 2020; Zhao et al., 2013) to analyse the CHN case in depth. Testing emissions with EMeRGe airborne BC and CO data would increase confidence in our past results using the ground-level data, to which transport/deposition errors for near-ground levels might have affected. Moreover, it was advantageous that CO$_2$ observation data relatively free of the effects of vegetation

(CO$_2$ uptake during photosynthesis and losses through respiration) were used to allow specific BC and CO emission factors for combustion to be calculated. A further objective was to identify residual ratios for pollutants emitted along with BC, CO, and CO$_2$ for the different regions by performing multi-species analysis.





The remainder of the manuscript contains Sect. 2, in which we explain the observational data, model configuration settings, and applied methods; Sect. 3, in which we compare the concentrations, emission estimates, and emission ratios; and Sect. 4,
in which we summarize the results and draw conclusions.

## 2 Material and methodology

### 2.1 Airborne observations in Asia during the EMeRGe campaign

This study is focused on analysing the combustion-related SLCFs, i.e. BC measured using a single particle soot photometer
(SP2) (by the Max Planck Institute for Chemistry in Mainz) (Holanda et al., 2020) and CO measured by UV photometry (by
the Deutches Zentrum für Luft- und Raumfahrt Institute of Atmospheric Physics) (Gerbig et al., 1996), and $CO_2$ measured by
cavity ring-down spectroscopy (by the Deutches Zentrum für Luft- und Raumfahrt Institute of Atmospheric Physics) (Chen et
al., 2010) onboard a HALO research aircraft during the EMeRGe campaign (https://www.iup.uni-bremen.de/emerge/home/;
Andrés Hernández et al., 2022). Pollution plumes from major population centres in Asia were detected during parts of flights
E-AS-03 (12 March 2018), E-AS-06 (20 March 2018), E-AS-08 (24 March 2018), E-AS-09 (26 March 2018), E-AS-10 (28
March 2018), and E-AS-13 (4 April 2018). These data were used to characterize emissions from near the surface to altitudes
of ~1–2 km. A custom merged dataset of 15 s (until flight E-AS-07) or 30 s (from flight E-AS-08) averages was prepared
according to the time resolution of the proton-transfer reaction mass spectrometry and analysed.

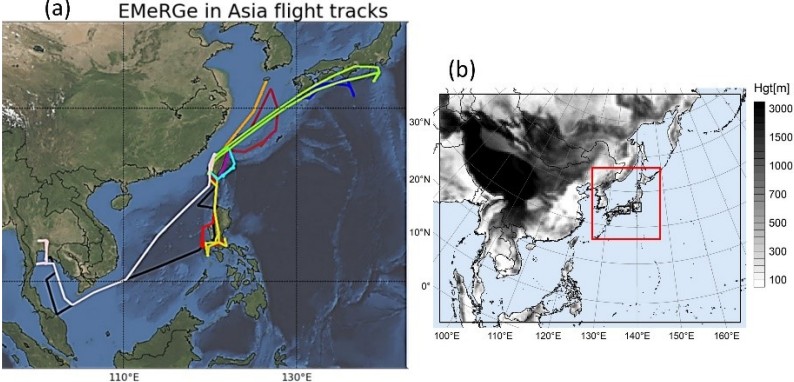

**Figure 1. (a) EMeRGe flight tracks in Asia: E-AS-03 (black), E-AS-04 (purple), E-AS-05 (brown), E-AS-06 (red), E-AS-08 ≅ E-AS-**
**09 (orange), E-AS-10 (yellow), E-AS-11 (blue), E-AS-12 (cyan), E-AS-13 (light green), E-AS-14 (white). (b) WRF/CMAQ model**
**target domains d01 (the whole figure) and d02 (the red box).**

### 2.2 WRF/CMAQ modelling system

The CMAQ modelling system version 5.0.2 (Byun and Schere, 2006) has been found to be suitable for accurately simulating
regional pollutant gas and particulate concentrations in Asia, including JPN. The model configuration is shown in Table 1
(Ikeda et al., 2014; Yamaji et al. 2020; Chatani et al., 2018, 2020). The CMAQ modelling system was coupled with the Weather



Research and Forecasting model (WRF) in the Advanced Research Weather Research and Forecasting (ARW) model version
3.7.1 (Skamarock et al., 2008), nudged by the US National Centers for Environmental Prediction Final Operational Model
Global Tropospheric Analyses (ds083.2) with a resolution of 1° × 1° (Gemmill et al., 2007), and also used real-time global sea
surface temperature high-resolution analysis with a resolution of 1/12° × 1/12° for the sea surface temperature. Two target
domains were considered in the CMAQ model, the East Asian area with a resolution of 45 km × 45 km (d01) and JPN with a
resolution of 15 km × 15 km (d02) (Figure 1b) extending vertically through 27 layers to 50 hPa. The initial concentrations
were the default conditions in the CMAQ model in the Statewide Air Pollution Research Center model version 07 (SAPRC07)
chemical category, for which calculations were performed more than a couple of months for the spin-up. Boundary
concentrations were the mean climatological data from the CHASER model (Sudo et al., 2002) using the Hemispheric
Transport of Air Pollution version 2.2 (HTAPv2.2) emission inventory (Huang et al., 2017; Janssens-Maenhout et al., 2015).
BC simulations were performed using the sixth-generation CMAQ aerosol module AERO6.

For domain d01 (East Asia excluding JPN), monthly anthropogenic emission data were taken from the HTAPv2.2 emission
inventory using 2010 as the reference year unless otherwise stated. The HTAPv2.2 emission inventory uses the mosaic Asian
anthropogenic emission inventory MIX version 1.1, which couples the Multi-resolution Emission Inventory for China
(MEICv1.0) developed by Tsinghua University (http://www.meicmodel.org) (Li et al., 2017), local emission inventories for
Korea and India, and the REASv2.1 for the rest of Asia (Kurokawa et al., 2013). The MEICv1.0 inventory provides month-to-
month gridded emission data with a spatial resolution of 0.25° for five emission sectors (http://meicmodel.org.cn/, last accessed
15 May 2023). The ratios of sectoral annual emissions in the MEICv1.0 were multiplied by factors selected to represent recent
changes in precursor emissions, as described by Zheng et al. (2018), and these data are later called the HTAPv2.2z emission
inventory. Open biomass burning emissions were taken from the Global Fire Emissions Database (GFEDv4.1s) as three-hourly
data (Giglio et al., 2013; Van der Werf et al., 2017). For domain d02 (JPN), anthropogenic emissions were taken from an
updated version of the Japan Clean Air Program and Japan Auto-Oil Program emission inventory database module (updated
JEI-DB; 2013) except that from vehicles (2010). Ship emissions were provided by the Sasakawa Peace Foundation at a
resolution of ~1 km × 1 km estimated from activities determined using an automatic identification system and averaged hourly
and for each day of the week. The JEI-DB and the ship emissions have contributed to progress in air quality modelling research
in JPN (Chatani et al., 2018, 2023; Li et al., 2017; Kurokawa et al., 2013; Shibata and Morikawa, 2021). Biogenic volatile
organic compound emissions were taken from the Model of Emissions of Gases and Aerosols from Nature as hourly data
(Guenther et al., 2012). The simulation period for domain d01 was 3 yr (2016–2018).

**Table 1. Configurations of the Weather Research and Forecasting / Community Multiscale Air Quality (WRF/CMAQ) modelling**
**system**

| **Chemistry Transport Model** | CMAQ5.0.2 | **Regional Meteorology Model** | WRF(ARW) v3.7.1 |
|---|---|---|---|
| **Boundary condition (d01)** | CHASER(MIROC-ESM v4.0) | **Nudging for WRF** | NCEP-FNL |



| Chemical Mechanism | SAPRC–07 | |
|---|---|---|
| Aerosol Module | AERO6 | |
| Domains and resolution | d01 (out of Japan): 45×45 km | d02 (within Japan): 15×15 km |
| Emissions | • Anthropogenic<br>  HTAPv2.2z     (month)<br>     REASv2.1 for the rest of Asia<br>     MEICv1.0 + Zheng et al. (2018)<br>     for China<br>     National inventories for South<br>     Korea and India<br>• Biomass burning<br>  GFEDv4.1s     (day, 3-hour)<br>• Biogenic VOCs<br>  MEGANv2.1   (day, hour)<br>• Volcano<br>  AeroCom (Diehl et al., 2021)<br>          (annual) | • Vehicles<br>  JEI-DB     (month,<br>  weekday/end, hour)<br>• Other anthropogenic<br>  JEI-DB (Updated)  (month, hour)<br>• Ship<br>  SPF     (month, day of week)<br>• Biomass burning<br>  GFEDv4.1s     (day, 3-hour)<br>• Biogenic VOCs<br>  MEGANv2.1   (day, hour)<br>• Volcano<br>  JMA Day (JMA online) (day) |

CHASER(MIROC-ESM) = Circulation Model for the Study of Atmospheric Environment and Radiative Forcing, NCEP-FNL = US National Centers for Environmental Prediction Final, SAPRC-07 = Statewide Air Pollution Research Center model version 07, AERO6 = sixth-generation CMAQ aerosol module, HTAP = Hemispheric Transport of Air Pollution emission inventory, JEI-DB = Japan Clean Air Program and Japan Auto-Oil Program emission inventory database module, REAS = Regional Emission Inventory for Asia, MEIC = Multi-resolution Emission Inventory for China, GFED = Global Fire Emissions Database, SPF = Sasakawa Peace Foundation, VOCs = volatile organic compounds, MEGANv2.1 = Model of Emissions of Gases and Aerosols from Nature, AeroCom = Aerosol Comparison between Observations and Models, JMA = Japan Meteorological Agency.

## 2.3 Backward trajectories and active fire data

We performed backward trajectory analyses using the US National Oceanic and Atmospheric Association Hybrid Single Particle Lagrangian Integrated Trajectory (HYSPLIT) 4 model (Draxler et al., 2018), which is an extensive system for computing simple air parcel trajectories (https://www.arl.noaa.gov/hysplit/, last accessed 30 May 2023), to determine the sources of pollutant emissions. Back trajectories for the investigated flight segments were drawn for 120 h (5 d) from individual three-dimensional locations at a resolution of 15–30 seconds as per the aircraft data averaging steps. The meteorological field



used in the HYSPLIT model was the Global Data Assimilation System with a resolution of 1° × 1°. More details about the HYSPLIT model were published by Stein et al. (2015).

Active fire data were used to reaffirm the type of emissions and the back-trajectory analysis results. Near real-time hotspot data were taken from the US NASA Fire Information for Resource Management System and were measured using the Visible Infrared Imaging Radiometer Suite (VIIRS) sensors on the Suomi National Polar-orbiting Partnership and US National Oceanic and Atmospheric Association 20 satellites (bright_ti5) (https://firms.modaps.eosdis.nasa.gov/map/, last accessed 30 May 2023).

**2.4 Residual ratios for co-emitted pollutants**

Flights for the EMeRGe-Asia campaign successfully sampled multiple sources of pollution in Asia, including in CHN and Southeast Asia. On these flights, spatiotemporal pollutant transport in the planetary boundary layer was assessed and residual ratios were determined. The residual ratio for two co-emitted pollutants, target species X and tracer species Y, defined as X/Y, is the ratio between the enhanced concentrations ($\Delta$) which calculated from their baseline levels. Temporal and spatial variations were found in the concentrations related to each source of emissions, therefore the residual ratios could be assessed by calculating the slopes of the regression lines fitted to scatter plots of the concentrations. The $BC/CO$, $CO/CO_2$, and $BC/CO_2$ ratios for the source regions are assessed in Sect. 3.2. The residual ratios for cases without rain along the air mass transport routes (Choi et al., 2020; Kanaya et al., 2020) were used to assess the sources of emission in various countries and changes in combustion technology.

$CO_2$ is an ideal reference gas for tracing fresh and aged anthropogenic emission plumes because it is a major product of combustion processes, although biogenic interactions make it difficult to separately determine $CO_2$ signals from combustion sources and strong surface sources and sinks (Takegawa et al., 2004). CO and $CO_2$ ratios have been used to determine the amounts of incomplete combustion occurring in developing and developed countries (Chandra et al., 2016; Palmer et al., 2006; Suntharalingam et al., 2004; Takegawa et al., 2004; Tang et al., 2018; Wang et al., 2010). BC and CO are byproducts of incomplete combustion of carbon-based fuels. Therefore, the ratio between BC and CO is a valuable indicator for characterizing combustion/emission sources and validating BC emissions in bottom-up inventories (Choi et al., 2020; Guo et al., 2017; Zhu et al., 2019; Kondo et al., 2006, 2011). $BC/CO$, $CO/CO_2$, and $BC/CO_2$ emission ratios differ among combustion sources, techniques, and environments (related to biogenic $CO_2$ sinks). Comparing observed with simulated concentration ratios and emission ratios in frequently used emissions inventories can identify the parts of an emissions inventory that require improvement. However, using residual ratios in plumes to indicate sources of pollution should be taken with care when assessing long-range transport of an air mass, particularly when the air mass was convectively transported to the lower stratosphere or over different underlying ecosystems (Yokelson et al., 2013).



## 2.5 Combined model–observation approach for estimating Chinese emissions

We used a combined model–observation approach to estimate emission correction factors for pollutants in the HTAPv2.2z emission inventory used in the CMAQ model. This was achieved by calculating observed-to-modelled concentration ratios by dividing the mean observed value by the mean simulated value for plumes. BC and CO emissions were estimated using a correction factor for BC (Eq. 1) and CO (Eq. 2), respectively, labelled $E(x)_{HTAPv2.2z}$ (or $E(x)$ for short), where x is BC or CO.

$$E(BC)_{HTAPv2.2z} = \text{mean observed [BC] / mean modelled [BC]} \qquad \text{(Eq. 1)}$$

$$E(CO)_{HTAPv2.2z} = \text{mean observed [}\Delta CO\text{] / mean modelled [}\Delta CO\text{]} \qquad \text{(Eq. 2)}$$

The observation-to-model concentration ratios are discussed in Sect. 3.1, and most deeply in 3.1.5, in which temporal changes in emissions in CHN are assessed using observed residual ratios to translate the BC emission rate into CO and then $CO_2$ emission rates. The BC, CO, and $CO_2$ emission rates were estimated using the relevant correction factors as linear emission-to-concentration (or concentration enhancement) ratios, assuming that the general geographic pattern in the HTAPv2.2z emission inventory was correct. The baseline BC concentrations were assumed to be zero, but the non-zero baseline for CO concentrations were considered (Kondo et al., 2011; Kanaya et al., 2016; 2020). A linear response was more valid for BC than CO because there were no known feedback processes linking the loss term to the source term for BC. A linear response is still a reasonable assumption for $\Delta CO$ (the enhancement from the baseline level) because emissions were the dominant causes of the increases and the losses were small at the time scale of air mass transportation. Several methods are available for estimating the CO baseline concentration (Matsui et al., 2011; Miyakawa et al., 2017; Oshima et al., 2012; Takegawa et al., 2004; Verma et al., 2011). For example, the 5th percentile of the 14 d data moving from the first day through the entire measurement period can be used (Choi et al., 2020; Kanaya et al., 2020). The CO baseline concentration used in Eq. 2 was calculated from the slope and intercept of the linear regression fitted to the BC/CO concentration scatter plot, which was described in Sect. 2.4.

The estimated amounts of BC and CO emitted (Est(x)) were calculated by multiplying E(BC) and E(CO), respectively, by the emission value assigned to CHN in the HTAPv2.2z emission inventory, as shown in Eq. 3. Estimated emissions of other substances (Est*(x)) were translated from the estimated amounts of BC and CO and the observed BC/CO and $CO/CO_2$ residual ratios (X/Y), as shown in Eqs. 4–6.

$$Est(x) = E(x)_{HTAPv2.2z} \times HTAPv2.2z(x), \qquad \text{(Eq. 3)}$$

where x is BC or CO.

$$Est^*(CO) = Est(BC) / \text{(observed BC/CO ratio)} \qquad \text{(Eq. 4)}$$

$$Est^*(BC) = Est(CO) \times \text{(observed BC/CO ratio)} \qquad \text{(Eq. 5)}$$

$$Est^*(CO_2) = Est(CO) / \text{(observed } CO/CO_2 \text{ ratio)} \qquad \text{(Eq. 6)}$$



## 3 Results and discussion

### 3.1 Evaluating BC, CO, and $CO_2$ concentrations in polluted cases and estimating emissions from CHN

The BC, CO, and $CO_2$ concentrations ($CO_2$ only from observations) determined from EMeRGe airborne observations and in CMAQ simulations, and backward trajectories from HYSPLIT model and distributions of hotspot data from VIIRS are shown in Figure 2 to Figure 5. The model performances for flight segments over the pollutant plumes were carefully analysed and divided into four cases, one for fire pollution (THL) and three for urban emissions from PHL, JPN, and CHN. No wet removal

process likely occurred (i.e., the relative humidity did not reach 100%) during the flight segments for these pollution events, and good correlations were found between the observed and simulated water concentrations (Fig. S1 in the supplement). The residual ratios were unchanged when a stricter analysis was performed using cases with zero accumulated precipitation along 72 h and 120 h trajectories (APT3 and APT5, respectively) in both observations and simulations (Fig. S6 in the supplement). Statistical parameters for the relationships between the observations and simulations for each case are shown in Table 2.

Emissions from commonly used bottom-up emission inventories were compared with the results of this study, and the data are shown in detail in Table S2 in the supplement.

### 3.1.1 Case study for THL (emissions from fires)

The pollutant peak for the THL case during flight E-AS-03 S1 (12 March 2018, 0517–0546 UTC) was caused by emissions from fires (Figure 2b, 2c, and 2d). The HYSPLIT model indicated that the sources were in Cambodia, Vietnam, and Myanmar,

and 9116 fire hotspots on 11 March were identified using the VIIRS. The highest concentrations at an altitude of 1.5 km were 3.26 μg m$^{-3}$ for BC (mean 1.63 μg m$^{-3}$, σ 0.80 μg m$^{-3}$), 580 ppbv for CO (mean 374 ppbv, σ 106 ppbv), and 420 ppm for $CO_2$ (Figure 2b and 2c). The model predicted similar maximum concentrations of 3.11 μg m$^{-3}$ for BC (mean 1.40 μg m$^{-3}$, σ 0.76 μg m$^{-3}$) and 722 ppbv for CO (mean 398 ppbv, σ 147 ppbv) during the flight, and the maxima were reached 10 min earlier than in the observations, which was attributed to limitations in the spatial and temporal distributions of the model inputs and

outputs. The simulated and observed maximum and mean concentrations were similar, but the correlation coefficients were affected by the time differences between the observations and predictions (Table 2 and Figure 6a and 4b). This is discussed further in Sect. 3.2.1 treating residual ratios. This agreement for the THL case indicated that the GFEDv4.1s inventory for fire emissions in Southeast Asia was suitable for use. This would have been because small fires associated with increased carbon and CO emissions are included in the GFEDv4.1s inventory compared to the GFEDv4.0 inventory (Van der Werf et al., 2017).

### 3.1.2 Case study for PHL (urban pollution)

Maximum BC and CO concentrations of 1.51 μg m$^{-3}$ and 241 ppbv, respectively, were found for flight segment E-AS-10 (0111–0419 UTC); a maximum CO concentration of 211 ppbv was found for flight E-AS-06 (0058–0347 UTC); and maximum BC and CO concentrations of 0.98 μg m$^{-3}$ and 197 ppbv, respectively, were found for flight E-AS-03 S2 (0905–0922 UTC) across PHL at altitudes of ~1–2 km (Figure 3b, 3c, 3e, 3f, 3i, and 3j). The primary source of BC and CO was the Manila urban



area (Figure 3d, 3g, and 3k). The BC and CO concentrations were approximately half of the concentrations for the THL case and were also lower than the concentrations for the CHN case (Sect. 3.1.4). CMAQ simulations underestimated the observed maxima, giving maxima of 0.25 and 0.29 μg m$^{-3}$ for BC for flights E-AS-10 and E-AS-03, respectively, which were biased by −23% to −30%, with maxima of 135–153 ppbv for CO (biased by −5% to −23%). The CMAQ simulation results were similar to the BC concentrations found in other global simulations (CAMchem–CESM2.1) and regional simulations

(WRFchem v4.3.3) for flight E-AS-10 (Deroubaix et al., 2024a). On the other hand, the IFS-CAMS simulation predicted the maximum CO concentration well in Deroubaix et al. (2024a), possibly because anthropogenic emissions in the IFS-CAMS simulation were taken from the CAMS-GLOB-ANTv4.2 emission inventory (Granier et al., 2019). The correlation coefficients for the relationships between the observations and our simulations were $R_{BC}$ = 0.49 and 0.54 and $R_{CO}$ = 0.51 and 0.59 (p-values ≤ 0.001) except for $R_{CO\_E-AS-03}$, which was 0.47 (p-values > 0.001) (Table 2). The observation/model ratios were 2.00±0.77

and 3.24±1.92 for BC and 1.07±0.55, 1.20±0.16, and 1.37±0.74 for CO (Figure 6c–6g), indicating that the REASv2.1 for PHL underestimated BC emissions more than CO emissions. This might have been related to emissions from the dominant residential sector being underestimated (contributing 76% and 63% of BC and CO emissions, respectively). Kurokawa et al. (2013) found that uncertainties in BC and CO emissions in REASv2.1 for Southeast Asian countries were ±257% and ±131%, respectively, mostly related to the residential sector (±351% for BC and ±208% for CO) because of insufficient information about emission factors and removal efficiencies and inadequate data for the road emissions sector. BC emissions in the

REASv2.1 were only half of the emissions given in the Emissions Database for Global Atmospheric Research inventory (EDGARv4.2), but CO amounts in both inventories were similar (Lee et al., 2018). Lee et al. (2018) found that the underestimated fine particulate (PM$_{2.5}$) mass concentration in the REASv2.1 relative to ground-based observations in Manila might have been related to certain aerosol sources (e.g., roads, construction, and industrial dust) being missing. The missing

sources of BC and CO for our study are likely combustion sources and may therefore be different from those discussed in Lee et al. (2018). Our CMAQ model simulations assumed emissions are unchanged since 2010 but the national carbon budget for PHL increased by 65% between 2010 and 2018 (Friedlingstein et al., 2020). BC, CO, and CO$_2$ emissions for 2015 were 13%, 14%, and 32% higher, respectively, in the newer version of the inventory (REASv3.2) (Kurokawa and Ohara, 2020) and in other bottom-up emission inventories, and these were probably more appropriate emissions for PHL, as discussed in Sect.

280  3.2.2.

### 3.1.3 Case study for JPN

For the JPN case, higher BC and CO concentrations were observed during flight E-AS-13 (4 April 2018) between approximately 0300 and 0600 UTC than between 0100 and 0300 UTC, and the CMAQ model predicted the concentrations between 0300 and 0600 UTC well (the dashed box in Figure 4b, 4c). However, the observed BC concentration between 0300

and 0600 UTC fluctuated, which caused the correlation between the observed and simulated BC concentrations to be relatively poor. We selected a sub-segment between 0428 and 0455 UTC (the black box in Figure 4b, 4c) at an altitude of ~2.5 km across



Sagami Bay and Suruga Bay (112 data points) (Figure 4a), where the observed CO and $CO_2$ concentrations correlated best ($R$ = 0.90). The maximum BC concentration was 0.37 µg m$^{-3}$ (mean 0.26±0.04 µg m$^{-3}$), and the back trajectory simulation indicated that the air mass had travelled over regions between Kagoshima and Osaka, Nagoya, and Tokyo (Figure 4d). For this

sub-segment, the CMAQ model underestimated the maximum BC concentration and gave a stable concentration of 0.13–0.17 µg m$^{-3}$ BC (mean 0.16±0.01 µg m$^{-3}$). The observations did not contain a clear maximum CO concentration (mean 130±7 ppbv); the concentration was slightly underestimated by the CMAQ model (mean 119±5 ppbv); and the correlation coefficients for the relationships between the observed and simulated BC and CO concentrations were low. Similar to the THL case, the poor correlations between the CMAQ model results and observed minute-scale variability were attributed to limitations in the

spatial and temporal distributions of the model inputs and outputs. As shown in Fig. S2b in the supplement, the model overestimated the BC concentrations below an altitude of ~1.5 km and underestimated the BC concentrations at altitudes of ~2–4 km over the Sagami Bay and Suruga Bay region (33°–35° N, 138°–140° E). Similar to JPN in this study, the CMAQ model (version 4.7) underestimated the BC concentrations over Fukue Island with air masses traveling over Japan's main islands in a previous study (Kanaya et al., 2020). It was difficult to determine whether the underestimation of the BC

concentration was related to the JEI-DB being used in this study. However, it should be noted that the JEI-DB has been found to underestimate the BC emission component of observations in Tokyo (Shibata and Morikawa, 2021).

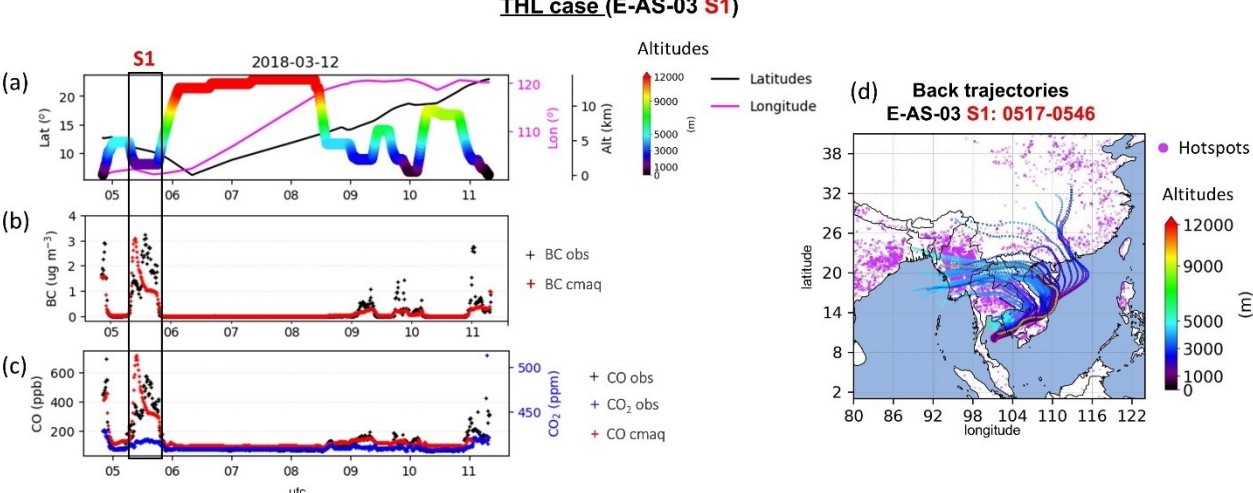

**Figure 2. Concentrations of black carbon (BC), CO, and CO₂ and HYSPLIT trajectories for the case of Thailand (THL). (a) Flight coordinates: latitudes (black) scaled on the left axis, longitudes (pink) scaled on the right axis, altitudes (colored) scaled on the second**
**right axis and colored-coded according to the color bar. (b) BC concentrations from observations (black) and simulations for domain d01 (45×45 km) (red). (c) CO and CO₂ concentrations: observed CO (black) and CO₂ (blue) and simulated CO (red). The S1 black box across (a), (b), and (c) indicate the flight segments selected for the THL case. (d) 120 h back trajectories of the flight segment in the THL case, colored-coded according to the color bar. In (d), the magenta dots are hotspots identified by the VIIRS. Each dot represents a single fire detected within an area of 760 m². The dot size was chosen for clarity only.**



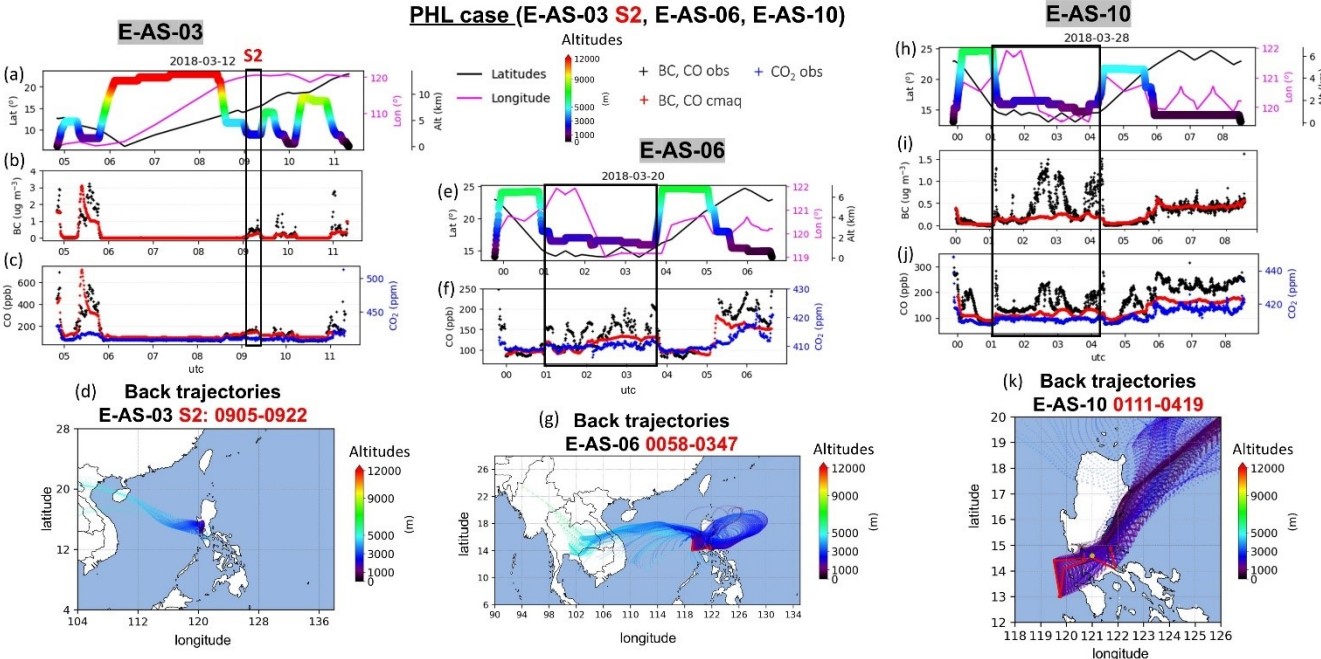


**Figure 3. Similar to Figure 2, but for the PHL case. (a), (e), and (h) show the coordinates of E-AS-03, E-AS-06, and E-AS-10, respectively, with legends similar to Figure 2 (a). (b) and (i) show the BC concentrations of E-AS-03 and E-AS-10, respectively, with legends similar to Figure 2 (b). (c), (f), (j) show CO and CO₂ concentrations of E-AS-03, E-AS-06, and E-AS-10, respectively, with legends similar to Figure 2 (c). (d) 120 h back trajectories of the flight segment in the PHL case, color coded according to the color**

**bar.**

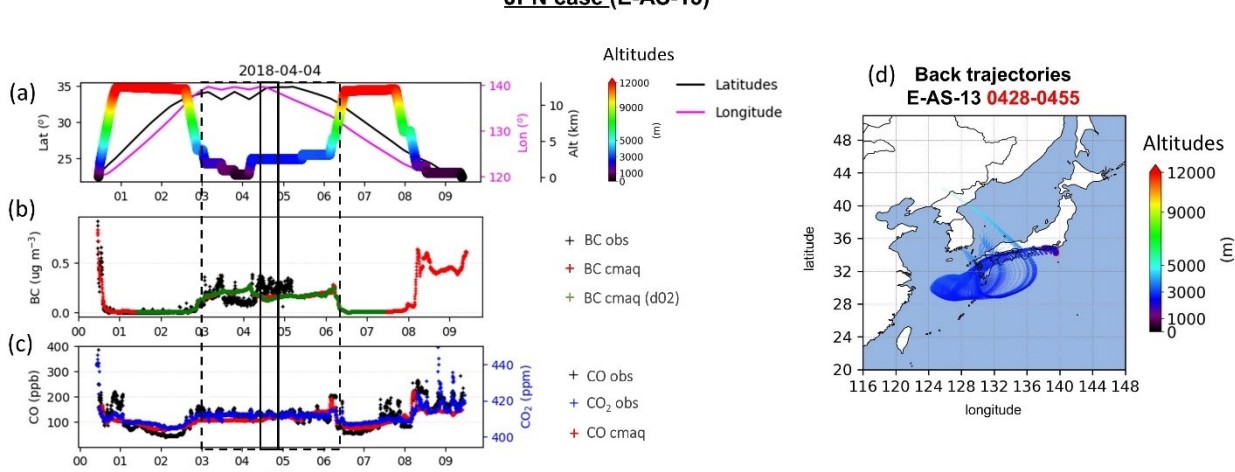

**Figure 4. Similar to Figure 2, but for the JPN case, except that (d) does not include hotspots as in Figure 2 (d). The dashed box over (a), (b), and (c) is to support the discussion in Sect. 3.1.3. The green crosses in (b) indicate simulation by domain d02 (15×15 km), which focuses on Japan.**





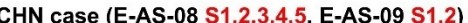

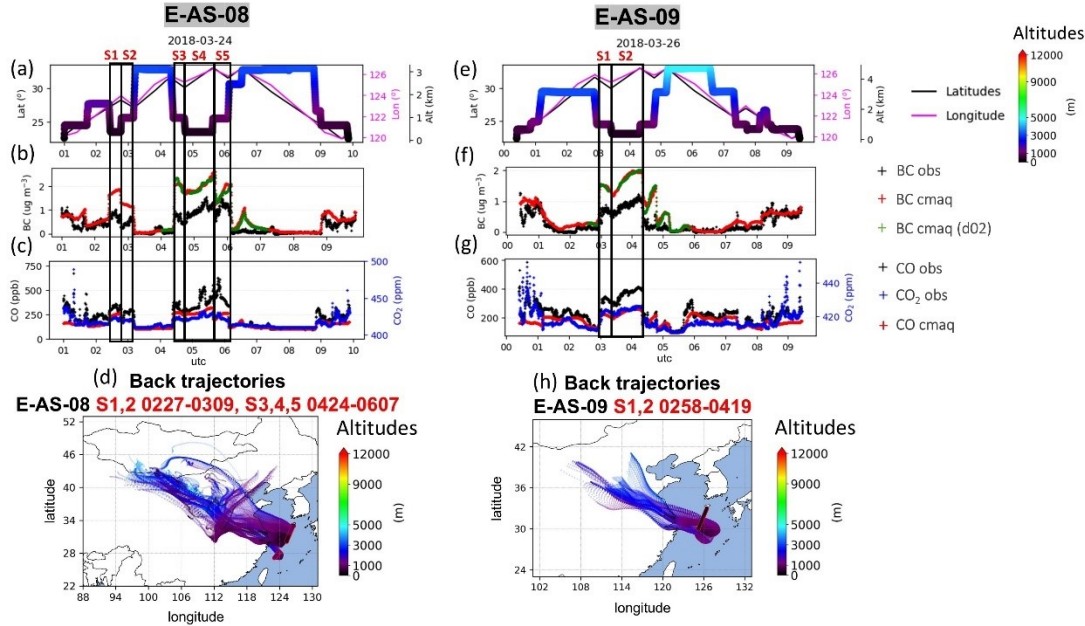

**Figure 5. Similar to Figure 4, but for the CHN case. (a) and (e) show the coordinates of E-AS-08 and E-AS-09, respectively, with legends similar to Figure 2 (a). (b) and (f) show the BC concentrations of E-AS-08 and E-AS-09, respectively, with legends similar to Figure 4 (b). (c) and (g) show the CO and CO₂ concentrations of E-AS-08 and E-AS-09, respectively, with legends similar to Figure 4 (c). (d) and (h) show 120 h back trajectories of the flight segments in the CHN case, colored-coded according to the color bar.**

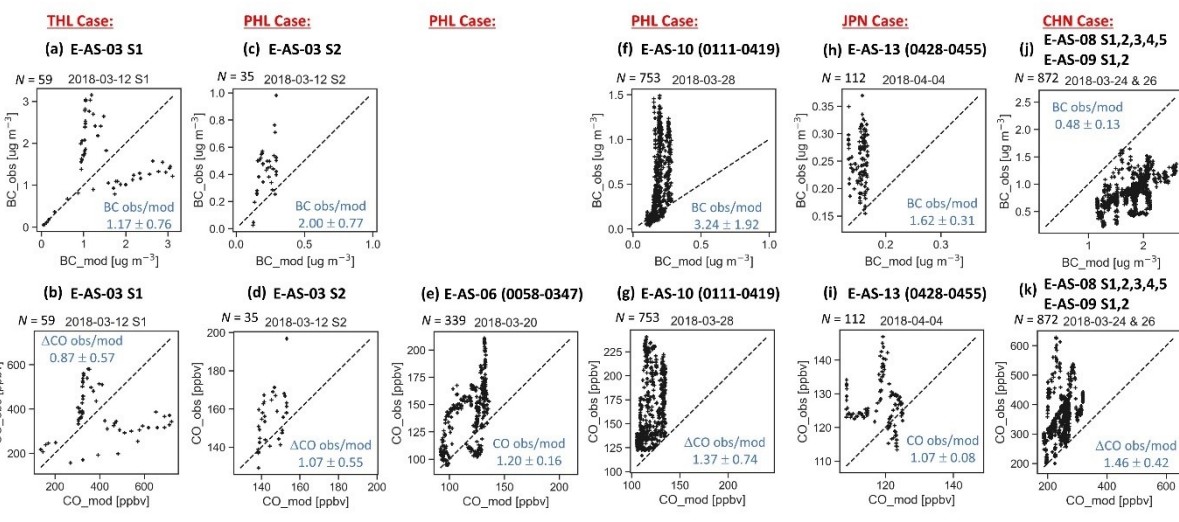

**Figure 6. Observed plotted against simulated black carbon (BC) concentrations (upper panels) and CO concentrations (lower panels). The dashed black lines are the 1:1 lines, and the observation/model ratios for BC and ΔCO and standard errors are shown in blue text. N is the number of data points. For (e and i), the observation/model ratio for CO instead of ΔCO is shown because a realistic CO baseline concentration was unavailable. Standard errors of full data in each case are shown for all panels except (a and b) flight E-AS-03 S1 averaged at every 2 minutes, (e) flight E-AS-06 averaged at every 5 minutes, and (f and g) flight E-AS-10 filtering with accumulated precipitation along 3 d trajectories ≤ 1 mm because calculations for standard errors are not valid for full data in those cases. Obs. = observed and Mod. = simulated.**





**Table 2. Statistical parameters (correlation coefficient _R_, number of data points _N_, _p_-values for correlations between observed (Obs.) and simulated (Mod.) data, maximum (Max.), mean, and standard deviation (σ)) for the BC and CO concentrations found for the examined cases (THL = Thailand, PHL = the Philippines, JPN = Japan, and CHN = China)**

| Case studies (Flight legs) | | THL case (E-AS-03 S1) | PHL case (E-AS-03 S2) | PHL case (E-AS-06) | PHL case (E-AS-10) | JPN case (E-AS-13) | CHN case (E-AS-08 S1–5 & E-AS-09 S1,2) |
|---|---|---|---|---|---|---|---|
| Time (UTC) | | 2018/3/20 0517-0546 (30-s) | 2018/3/12 0905-0922 (30-s) | 2018/3/20 0058-0347 (30-s) | 2018/3/28 0111-0419 (15-s) | 2018/4/4 0428-0455 (15-s) | 2018/3/24,26 Table S1 (15-s) |
| BC | _R_ | -0.01 | 0.49 | / | 0.54 | 0.01 | 0.74 |
| | _N_ | 60 | 35 | / | 753 | 112 | 872 |
| | _P_ | > 0.05 | > 0.001 | / | < 0.001 | > 0.05 | < 0.001 |
| _Max._ _(Mean±σ)_ _(μg m⁻³)_ | _Obs._ | 3.11 (1.63±0.80) | 0.98 (0.44±0.18) | | 1.49 (0.48±0.38) | 0.37 (0.26±0.04) | 1.66 (0.84±0.27) |
| | _Mod._ | 3.26 (1.40±0.76) | 0.29 (0.22±0.06) | / | 0.28 (0.18±0.04) | 0.17 (0.16±0.01) | 2.60 (1.77±0.33) |
| CO | _R_ | -0.08 | 0.47 | 0.59 | 0.51 | 0.07 | 0.58 |
| | _N_ | 60 | 35 | 339 | 747 | 112 | 872 |
| | _P_ | > 0.05 | > 0.001 | < 0.001 | < 0.001 | > 0.05 | < 0.001 |
| _Max._ _(Mean±σ)_ _(ppbv)_ | _Obs._ | 580 (374±106) | 197 (155±13) | 211 (142±26) | 241 (157±34) | 149 (130±7) | 628 (351±69) |
| | _Mod._ | 722 (398±147) | 153 (145±6) | 136 (119±14) | 135 (117±9) | 125 (119±5) | 320 (246±31) |

### 3.1.4 Air masses from urban areas in CHN

The air masses with maximum BC and CO concentrations near Taiwan, Jeju (South Korea), and Okinawa (JPN) during flights E-AS-08 (24 March 2018, 0227–0309 UTC and 0424–0607 UTC) and E-AS-09 (26 March 2018, 0258–0419 UTC) gave a total of 872 data points and originated in Central East China (CEC) (Figure 5). These air masses were not affected largely by wet removal processes (APT3 max=1.3 mm, mean=0.1 mm, σ=0.3 mm). The maximum concentrations of BC and CO were 1.66 μg m⁻³ and 628 ppbv, respectively, and the $CO_2$ concentration increased from 419 to 433 ppm. The overall correlation coefficients for the relationships between the modelled and observed concentrations were moderate ($R_{BC} = 0.74$ and $R_{CO} = 0.58$, p-values < 0.001; Table 2). The model overestimated the BC concentration and underestimated the CO concentration by maxima of +1.62 μg m⁻³ and −400 ppbv, respectively, and the maximum simulated concentrations were 2.60 μg m⁻³ for BC and 320 ppbv for CO (Figure 5b, 5c, 5f, and 5g). Similar results were reported by Kanaya et al. (2020), who found the same tendencies in the CMAQv4.7.1 model using the REASv2.1 for BC and CO with respect to observed concentrations at Fukue Island (JPN), including for the EMeRGe flight E-AS-08 and E-AS-09 periods. The observation/simulation ratios (Eqs. 1 and 2) were 0.48±0.13 for BC and 1.46±0.42 for ΔCO (Figure 6j and 6k). The HTAPv2.2z emission inventory reflected recent decreases in Chinese emissions (Chatani et al., 2020; Zheng et al., 2018), but these updates might still be insufficient for accurately predicting BC concentrations.



In more detail, seven sub-air masses in the CHN case were separately investigated: One group was from southern CEC (S-CEC) and contained flight E-AS-08 S1–S3 and flight E-AS-09 S2, and another group was from both northern and southern CEC (NS-CEC) and contained flight E-AS-08 S4–S5 and E-AS-09 S1. Details about the sub-air-masses are shown in Table S1 in the supplement. The S-CEC sub-air-mass had a mean BC concentration of 0.73 µg m$^{-3}$ and a mean enhanced CO concentration ΔCO of 181 ppbv. The NS-CEC sub-air-mass had a mean BC concentration of 0.85 µg m$^{-3}$ and a mean CO enhanced concentration ΔCO of 338 ppbv. The ΔCO concentrations were calculated using the sub-air-mass baseline CO concentration using a function of the intercept and slope for the linear equations fitted to the BC/CO ratios (see Sects. 2.4 and 2.5). The observations made at Fukue Island (Kanaya et al., 2020) contained two similar peaks from northern CEC (N-CEC) and S-CEC on 24–26 and 26–28 March, respectively. Our mean results for the S-CEC air masses (0.73 µg m$^{-3}$ for BC and 181 ppbv for ΔCO) were similar to the mean results found by Kanaya et al. (2020) (0.67 µg m$^{-3}$ for BC and 190 ppbv for ΔCO), and the maximum concentrations found for the NS-CEC air mass in our study (0.85 µg m$^{-3}$ for BC and 338 ppbv for ΔCO) were similar to the maximum concentrations found for the N-CEC air mass by Kanaya et al. (2020) (0.92 µg m$^{-3}$ for BC and 254 ppbv for ΔCO) (**Error! Reference source not found.** in the supplement). Further comparisons of aircraft and ground-based observations are discussed in Sect. 3.2.4. The biases in the modelled concentrations relative to the observed concentrations were largest for flight E-AS-08 S3 and S5 sub-air-masses (+1.18 µg m$^{-3}$ for BC and −364 ppbv for CO, respectively), both at an altitude of ~900 m.

### 3.1.5 Estimated BC, CO, and CO$_2$ emissions from CHN

The overall process used to determine BC, CO, and CO$_2$ emissions in CHN is shown in Figure 7 and the results are shown in Figure 8 along with the results of other bottom-up inventories. The differences between our estimated emissions and other inventories are shown in Table S2 in the supplement. The degrees of consistency in the BC, CO, and CO$_2$ emissions are shown as a triangle plot in Figure 9.

The combined model–observation approach (Sect. 2.5) was applied to the Chinese air masses, and the calculated emission correction factor E(BC)$_{HTAPv2.2z}$ was 0.48±0.13 (Figure 7a). The mean ΔCO concentrations in the observed and simulated data were 245 and 168 ppb, respectively, giving a correction factor E(CO)$_{HTAPv2.2z}$ of 1.46±0.42 (Figure 7b). The objective was to compare the results using an approach similar to that used for ground-based observations at Fukue Island for the duration of the EMeRGe flights (Kanaya et al., 2020). Chinese emissions in the HTAPv2.2z emission inventory used for Eq. 4 were 1.36 Tg yr$^{-1}$ for BC and 134 Tg yr$^{-1}$ for CO (Figure 7). Emission rates for CHN were calculated using the spring 2018 mean BC/CO concentration ratio of 4.9±0.1 ng m$^{-3}$ ppb$^{-1}$ from observations made at Fukue Island (Kanaya et al., 2020) and the airborne CO/CO$_2$ concentration ratio of 21.1±0.4 ppb ppm$^{-1}$ (Figure 7).

The E(BC)$_{HTAPv2.2z}$ value was used as a basis for estimating emissions, and the results and uncertainty ranges are shown in Figure 8 as red triangles and red boxes, respectively, to allow them to be compared with the results and uncertainty ranges found in previous studies. The calculated BC emission rate was 0.65±0.25 Tg yr$^{-1}$, which was ~52% lower than the emission



rate of 1.36 (Tg BC) yr$^{-1}$ in the HTAPv2.2z emission inventory used in the CMAQ model (this study, marked with a black cross in Figure 8a). The method used to estimate the uncertainties is discussed below. Similar to recent estimates made using

observations at Fukue Island and Noto (Kanaya et al., 2020; Ikeda et al., 2022, 2023; Miyakawa et al., 2023), we found lower BC emission rates than are given in bottom-up emission inventories. Our BC emission rate was 38% lower than the emission rate estimated using long-term surface observations made in 2009–2018 at Fukue Island (1.06±0.29 (Tg BC) yr$^{-1}$; Kanaya et al., 2020), shown as green lines in Figure 8a. This may partly have been caused by different Chinese air masses being selected. The ECLIPSEv6b BC emission rate (0.96 Tg yr$^{-1}$, shown as dark grey squares in Figure 8a) was closest to the estimated

uncertainty range (0.4–0.9 Tg yr$^{-1}$, shown as a red box in Figure 8a) because it took into account the changes made in CHN to improve air quality as part of the 12th 5–year plan (Klimont et al. 2017). Note that the BC emission rate in CEDS (CMIP6) (2.54 Tg yr$^{-1}$ for 2014, shown as a blue line with squares in Figure 8a) much decreased in the newer version CEDS v_2021_02_05 (1.58 Tg yr$^{-1}$ for 2014 and 1.22 Tg yr$^{-1}$ for 2018, shown as an orange line with squares in Figure 8a). This would have been because updated International Energy Agency's energy data with less biomass consumed after 2000 and

improving industrial emissions of BC and OC in CHN were used to calculate the CEDS v_2021_02_05 BC emission rate (O'Rourke et al., 2021). However, the CEDS v_2021_02_05 BC emission rate was still higher than our estimate. The estimated CO emission rate was 166±65 Tg yr$^{-1}$, which was ~24% higher than the CO emission rate given by the HTAPv2.2z emission inventory of 134 Tg yr$^{-1}$ (this study, marked with a black cross in Figure 8b). Our estimated CO emission rate was 10%–32% higher than the highest (CEDS v_2021_02_05) CO emission rate of 150 Tg yr$^{-1}$ and the lowest (EDGARv6.1) CO emission

rate of 114 Tg yr$^{-1}$ among other emission inventories, which are shown as orange and black lines with squares, respectively, in Figure 8b. Note that the CEDS v_2021_02_05 and EDGARv6.1 CO emission rates were calculated using available activity data and emission factors without any observation-based corrections (Crippa et al., 2018). The CO_TCR2 is the CO emission rate calculated from top-down estimates (shown as a red line with stars in Figure 8b) was close to our estimate of 153 Tg yr$^{-1}$ for 2019–2020. Our estimated CO$_2$ emission rate was (12.4±4.8)×10$^3$ Tg yr$^{-1}$, which was similar to the EDGARv6.1 CO$_2$

emission rate (11.5×10$^3$ Tg yr$^{-1}$) and CEDS v_2021_02_05 CO$_2$ emission rate (10.2×10$^3$ Tg yr$^{-1}$), shown as black and orange lines with squares in Figure 8c, respectively, taking the uncertainty range into consideration. All of the investigated emission inventories indicated that CO$_2$ emissions increased between 2010 and 2020 and gave higher CO$_2$ emission rates than the reported carbon budget for CHN for 2018 (9.9×10$^3$ Tg yr$^{-1}$, shown as black dashes in Figure 8c) (Friedlingstein et al., 2020). We could alternatively use E(CO)$_{HTAPv2.2z}$ as a starting point for estimating emissions (Figure 7b). The results and uncertainty

ranges are shown as green triangles and green boxes in Figure 8. The CO emission rate estimated using E(CO)$_{HTAPv2.2z}$ was 195±59 Tg yr$^{-1}$, which was 46% higher than the CO emission rate in the HTAPv2.2z emission inventory but close to the uncertainty range (shown as a black cross and a green box in Figure 8b). Only the CEDS v_2021_02_05 CO emission rate fell within our uncertainty range (shown as an orange line with squares in Figure 8b). This method gave a higher estimated Chinese BC emission rate but a lower uncertainty, 0.77±0.23 Tg yr$^{-1}$, than the method using E(BC), due to lower errors from instrument

and model-specific transport representation as discussed later. Even so, BC emission rate from this method was still 44% lower



than the HTAPv2.2z emission inventory BC emission rate (shown as a black cross in Figure 8a). The estimated Chinese $CO_2$ emission rate was $(14.5\pm4.4)\times10^3$ Tg yr$^{-1}$, and the uncertainty range covered the EDGARv6.1 $CO_2$ emission rate $(11.5\times10^3$ Tg yr$^{-1}$) and CEDS v_2021_02_05 $CO_2$ emission rate $(10.2\times10^3$ Tg yr$^{-1}$), shown as black and orange lines with squares in Figure 8c. Our estimated Chinese $CO_2$ emission rate was high relative to emission rates calculated in other studies. An extra

7% $CO_2$ emissions caused by humans and livestock breathing in CHN (Cai et al., 2022) is usually not counted in bottom-up inventories but could partially explain the high $CO_2$ emission rate estimated using the E(CO) method. The E(CO)-based emission estimates were worse than the E(BC)-based emission estimates relative to the CO and $CO_2$ emission reference ranges. This multi-species analysis corroborated the conclusion that the HTAPv2.2z (this study), EDGARv6.1, and CEDS v_2021_02_05 emission inventories require BC and CO emissions in CHN to be corrected downward and upward, respectively.

The uncertainties of the emission estimates were propagated from (1) the uncertainties in our observation-to-model ratios and emission ratios for multiple species (~26% for the E(BC)-based method, including standard errors for the data) and (2) the systematic errors of the instrument (10% for the single particle soot photometer (Ohata et al., 2021)). We estimated (3) representation errors caused by the limited opportunities for aircraft observations to be made in terms of seasonal and spatial variabilities of ±4% by comparing the spring and annual data, CEC region, and large footprint for CHN found at Fukue Island

(32.75° N, 128.68° E), assuming that BC emissions from CEC are the dominant source of BC (69.7%) (Fig. S5 in the supplement and Kanaya et al., 2020). Analysis of the CMAQ model and ground-based data at Fukue Island for the CEC air mass for 24–28 March gave a similar correction factor for BC emissions (E(BC) = 0.44) to correction factors found for the aircraft data, indicating that the ratios for the Chinese air mass found for the CMAQ (HTAPv2.2z) emission inventory were robust. We estimated (4) an error for model-specific transport of 26%, which was determined by comparing the BC

concentrations simulated by the CMAQ model with BC concentrations simulated by the Comprehensive Air Quality Model with eXtensions (CAMx) system using the same emission data (Chatani et al., 2018) for the model comparison framework (Itahashi et al., 2018; Yamaji et al., 2020). Overall, the estimated uncertainties in the E(BC)-based emission rates were ~39% (0.25 Tg yr$^{-1}$ for BC, 65 Tg yr$^{-1}$ for CO, and $4.8\times10^3$ Tg yr$^{-1}$ for $CO_2$, as stated above). The uncertainty for BC emission in this study (±0.25 Tg yr$^{-1}$) was similar to the uncertainty of the estimate using surface observations at Fukue Island (±0.29 Tg

yr$^{-1}$), although they studied for longer period (2009–2018) (Kanaya et al., 2020).

For emissions estimated using E(CO), the error term (1) was ~28%, instrument errors (2) was 4% for the UV photometer (Gerbig et al., 1996) and transport errors (4) were 7%, giving overall uncertainties of ~30% (0.23 Tg yr$^{-1}$ for BC, 59 Tg yr$^{-1}$ for CO, and $(4.4\times10^3)$ Tg yr$^{-1}$ for $CO_2$).

As shown in Figure 9, the degree of consistency between the BC, CO, and $CO_2$ emission rates (shown as orange triangles for

the E(BC) method and green triangles for the E(CO) method) could be assessed by comparing the estimate uncertainties (shown as orange and green lines with round marks for the E(BC) method and E(CO) method, respectively) with the CO and $CO_2$ reference emissions (113–150 (Tg CO) yr$^{-1}$ and $(9.9–11.5)\times10^3$ (Tg $CO_2$) yr$^{-1}$, shown as red lines with round marks). The CO and $CO_2$ emissions calculated using the E(BC) method (orange) were closer to the reference ranges (red lines), but the BC emission rates were lower and more different from the BC reference range (1.11–1.29 (Tg BC) yr$^{-1}$) (a reddish orange line on



the BC axis) than the CO, CO₂, and BC emission rates calculated using the E(CO) method (green). Uncertainties in CO₂ emission rates are much lower than uncertainties in BC and CO emission rates in emission inventories, and the good agreement between the CO/CO₂ ratios calculated from the observations and emission inventories in this study (discussed in Sect. 3.2.4) indicated that the CO emission rate was estimated more accurately than the BC emission rate in the bottom-up inventories. Therefore, the E(BC) estimation method was better than the E(CO) method because the calculated CO and CO₂ annual

emissions were more consistent with annual emissions in the bottom-up inventories. Calculating E(CO) value was associated with larger errors than calculating E(BC) value, i.e. standard errors were 31% vs 26% (Figure 7). This error (31%) didn't take into account the variations in the CO baseline concentration, which was calculated using BC/CO ratio equations affected by the selected sub-air-masses. For example, large and unrealistic E(CO) value differences were found for the NS-CEC and S-CEC sub-air-masses (1.93 and 1.29, respectively). Our BC emission estimates, including uncertainty ranges, were lower than

estimated in other emission inventories (Figure 9, shown by orange and green lines compared to the red line on the BC axis). BC emissions in the HTAPv2.2z emission inventory would need to be decreased by 45%–54% and BC emissions in the EDGARv6.1, HATPv3, and CEDS v_2021_02_05 inventories would need to be decreased by 31%–49% to match our estimates. Lower BC emission rates would be more consistent with bottom-up-calculated CO₂ emissions. This was confirmed for the first time. CO emissions in the HTAPv2.2z emission inventory would need to be increased by 24%–46% and CO

emissions in the EDGARv6.1, HTAPv3, and CEDS v_2021_02_05 inventories would need to be increased by 10%–42% to match our estimates (Table S2 in the supplement).

In conclusion, multi-species analysis suggested that bottom-up emission estimates for CHN in the HTAPv2.2z emission inventory need to be decreased by ~50% for BC and increased by ~20% for CO to minimize the differences from measured data based on the E(BC) method.

(a) Emission correction factor: $E(BC)_{HTAPv2.2z}$ (Eq. 1)
= 0.48±0.13

(Eq. 3) | $HTAPv2.2z(BC)$, Tg yr⁻¹
= 1.36

Estimated BC emission: Est(BC), Tg yr⁻¹
0.65±0.25

(Eq. 4) | $S_o(BC/CO)$, ng m⁻³ ppb⁻¹
= 4.90±0.11

Estimated CO emission: Est*(CO), Tg yr⁻¹
166±65

(Eq. 6) | $S_o(CO/CO_2)$, ppb ppm⁻¹
= 21.1±0.4

Estimated CO₂ emission: Est*(CO₂), Tg yr⁻¹
(12.4±4.8) × 10³

(b) Emission correction factor: $E(CO)_{HTAPv2.2z}$ (Eq. 2)
=1.46±0.42

$HTAPv2.2z(CO)$, Tg yr⁻¹
= 133.85 | (Eq. 3)

Estimated CO emission: Est(CO), Tg yr⁻¹
195±59

$S_o(CO/CO_2)$, ppb ppm⁻¹ (Eq. 6)   (Eq. 5) $S_o(BC/CO)$, ng m⁻³ ppb⁻¹
= 21.1±0.4                                      = 4.90±0.11

Estimated CO₂ emission :          Estimated BC emission :
Est*(CO₂), Tg yr⁻¹                Est*(BC), Tg yr⁻¹
(14.5±4.4) × 10³                  0.77±0.23






**Figure 7. Combined model–observation approach used to estimate black carbon (BC), CO, and CO₂ emissions using emission correction factors for (a) BC and (b) CO applied to the aircraft-based observation with regards to the HTAPv2.2z emission inventory**

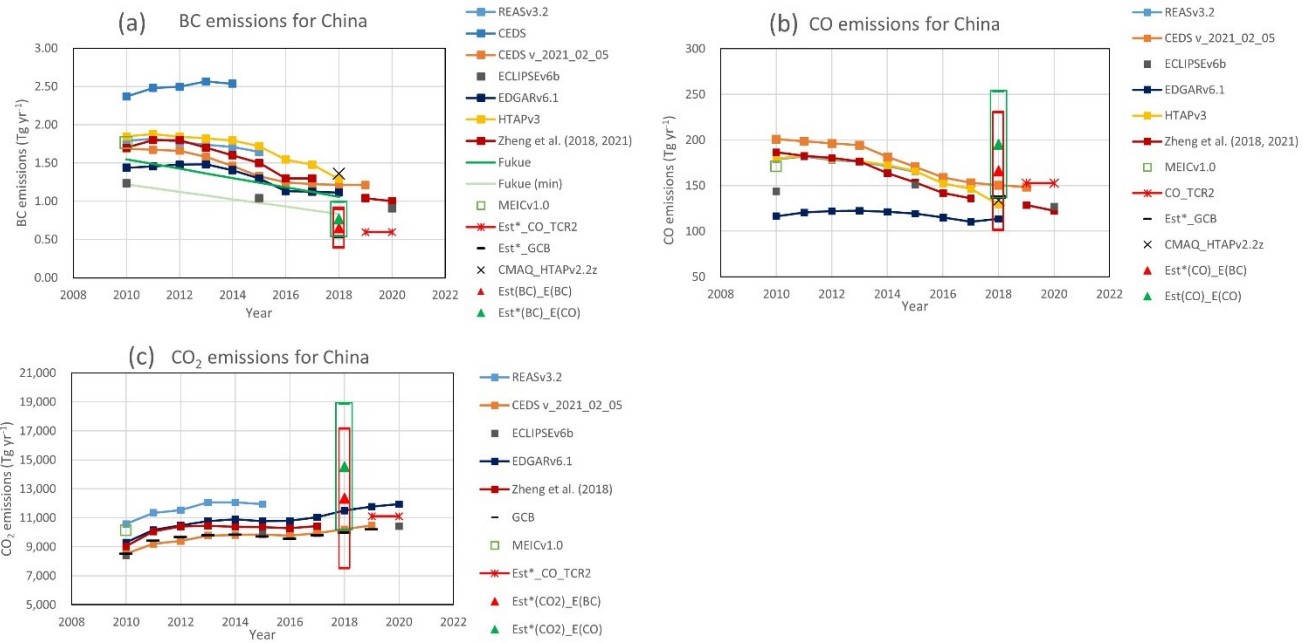

**Figure 8. Chinese BC, CO, and CO₂ emissions. The estimates made using the E(BC)<sub>HTAPv2.2z</sub>-based and E(CO)<sub>HTAPv2.2z</sub>-based methods**
**are shown as solid red and green triangles, respectively, with red and green boxes indicating the uncertainty ranges. Values from commonly used bottom-up emission inventories and other studies are shown, including from the Multi-resolution Emission Inventory for China (MEIC) v1.0 (open green squares), the Hemispheric Transport of Air Pollution (HTAP) v2.2z emission inventory (black batches), the global carbon budget (Friedlingstein et al., 2020) (black dashes), the CO TCR2 (Miyazaki et al., 2020) (red lines with stars), and the BC emission estimate at Fukue Island (green) and the minimum range (light green) (Kanaya et al.,**
**2020). Other references are given in Table S2 in the supplement.**



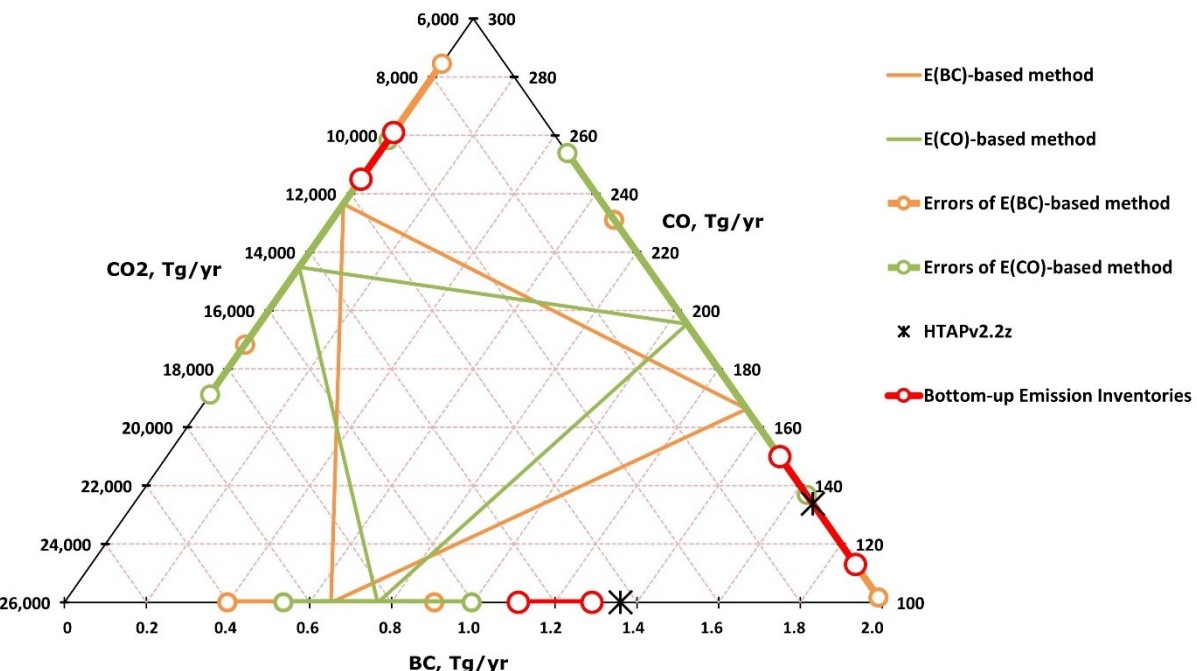

**Figure 9. Cross-species consistency checking of the estimated BC, CO, and $CO_2$ emissions. The orange and green triangles are estimates made using the $E(BC)_{HTAPv2.2z}$-based and $E(CO)_{HTAPv2.2z}$-based methods, respectively. Uncertainty ranges for these estimations are shown as orange and green lines with round marks as shown in the legend. Red lines with round marks are the reference emissions from bottom-up emission inventories. Black stars are emissions in the Hemispheric Transport of Air Pollution (HTAP) v2.2z emission inventory.**

### 3.2 Evaluation of emission ratios from bottom-up inventories using observed residual ratios

The residual ratios (slopes) between BC, CO, and $CO_2$ concentrations recorded during the EMeRGe flights and calculated in various commonly used bottom-up inventories for the selected Asian countries/regions are compared here. We will add context from the past decade to assess the accuracy of the emission ratios for co-emitted pollutants from different sources.

In Figure 10, the slopes are shown for BC and CO from observation and simulation of CMAQ. Also shown are slopes for CO and $CO_2$, and BC and $CO_2$ from observations. Temporal changes in the BC/CO, CO/CO₂, and BC/CO₂ emission and concentration ratios for the PHL, JPN, and CHN cases are shown in Figure 11. It can be seen from Figure 11 that the CO/CO₂ and BC/CO₂ emission ratios decreased in the PHL, JPN, and CHN cases over the past decade but that there was no clear trend in the BC/CO ratios except for in the EDGARv6.1 inventory.

### 3.2.1 THL case

The maxima in the THL case (caused by natural fires) were $7.1\pm0.3$ ng m$^{-3}$ ppb$^{-1}$ for the BC/CO ratio, $39.5\pm4.1$ ppb ppm$^{-1}$ (~4%) for the CO/CO₂ ratio, and $281\pm33$ ng m$^{-3}$ ppm$^{-1}$ for the BC/CO₂ ratio, and the correlation coefficients were $R_{BC/CO} = 0.95$, $R_{CO/CO2} = 0.78$, and $R_{BC/CO2} = 0.74$ ($p$-values < 0.0001) (Figure 10a–10c), indicating that these pollutants had one source.



The CMAQ model gave a similarly high BC/CO ratio of 5.20±0.04 ng m$^{-3}$ ppb$^{-1}$ and a correlation coefficient $R_{BC/CO} > 0.99$. The spatial mean BC/CO emission ratio for the box domain covering the THL case (13°–18° N, 100°–113° E, shown as a sparsely dashed box in Fig. S5 in the supplement) was 8.2 ng m$^{-3}$ ppb$^{-1}$ for all of the emission sectors in the CMAQ model and 7.3 ng m$^{-3}$ ppb$^{-1}$ using the GFEDv4.1s inventory, so was generally consistent with the BC/CO emission ratio calculated from observations. The simulated BC/CO ratio (5.2 ng m$^{-3}$ ppb$^{-1}$) was lower than the ratio calculated from observations (7.1

ng m$^{-3}$ ppb$^{-1}$), corresponding to the CMAQ model giving a higher maximum CO concentration than was observed (Figure 2c). This might have been caused by anthropogenic sources of CO in Thailand, which gave low BC/CO emission ratios in the model (4–6 ng m$^{-3}$ ppb$^{-1}$ as shown in Fig. S5d in the supplement). The BC/CO emission ratio calculated from the aircraft-based measurements was also similar to ratios calculated in previous studies, including from an emission factor database (Akagi et al., 2011), when simulated using the WRF-Chem (FINNv1.5) system (Lee et al., 2018), from previous airborne

observations (Kondo et al., 2011; Warneke et al., 2009), and from observations at a ground-based station with biomass-burning influence (Zhu et al., 2019) (Table S5 in the supplement). Very high and low BC/CO ratios have also been found in previous studies (Chi et al., 2013, Cristofanelli et al., 2013; Kondo et al., 2011; Paris et al., 2009; Vasileva et al. 2017; Warneke et al., 2009), but higher values for Asia than North America are linked to higher modified combustion efficiencies, inversely proportional to the $\Delta CO/\Delta CO_2$ ratios, during the flame phase of combustion (Kondo et al., 2011). BC/CO ratios tend to be

lower and CO/CO$_2$ ratios tend to be higher in the smouldering phase (higher CO emissions), but BC and CO$_2$ emissions tend to be higher in the flame phase (Laursen et al., 1992; Ward et al., 1992; Nance et al., 1993; Le Canut et al., 1996; Yokelson et al., 1999; Goode et al., 2000). However, different CO/CO$_2$ ranges for different types of fire might not be explicit from observations (Table S5 in the supplement). The BC/CO ratios calculated from the studied aircraft-based measurements and the low CO/CO$_2$ ratio for fires (~4%) indicate that the fires in the THL case might be dominated by very fresh spring Asian savanna

flaming fires, consistent with the GFEDv4.1s fire type classification for this region (Andela et al., 2019; Van der Werf et al., 2017).

### 3.2.2 PHL case

The CO$_2$ concentrations during flights E-AS-03 S2 and E-AS-10, during the maxima caused by emissions from the Manila area, did not positively correlate with the BC and CO concentrations, possibly because of large variations in CO$_2$ concentrations

caused by biogenic fluxes (Figure 3b, 3c, 3f, 3i, and 3j). However, other combinations are of interest. The observed BC/CO ratios converged around 11 ng m$^{-3}$ ppb$^{-1}$ (11.4±1.3 ng m$^{-3}$ ppb$^{-1}$ for flight E-AS-03 ($R = 0.83$) and 11±0.1 ng m$^{-3}$ ppb$^{-1}$ for flight E-AS-10 ($R = 0.97$)), which was the highest for the cases we investigated, indicating that a higher proportion of BC relative to CO was emitted from PHL than from the other urban areas we investigated. The BC/CO ratio simulated using the CMAQ model for flight E-AS-03 S2 (10.2±0.4 ng m$^{-3}$ ppb$^{-1}$, $R = 0.97$) agreed reasonably well with the observed ratio,

indicating that the BC/CO emission ratio calculated using the REASv2.1 was appropriate for the Manila region. In contrast, the simulated BC/CO ratio for flight E-AS-10 (4.3±0.1 ng m$^{-3}$ ppb$^{-1}$, $R = 0.91$) underestimated the observed ratio because the

model underestimated the BC and CO concentrations to different degrees. The BC/CO emission ratio for PHL calculated using the CMAQ (REASv2.1) model was 8.0 ng m$^{-3}$ ppb$^{-1}$, which still underestimated the observed ratio by ~30% (Figure 11a). The low simulated BC/CO ratio for flight E-AS-10 was difficult to explain but may have resulted from mixing of multiple sources, suspected from an unusually low CO intercept (~ 75 ppb on x-axis, Figure 10e) of the regression line. The observed CO/CO$_2$ ratio for flight E-AS-06 was 21.2±1.1 ppb ppm$^{-1}$, which was similar to the ratio for urban air masses for the CHN case but 34% lower than the lowest ratio for the bottom-up inventories (28 ppb ppm$^{-1}$, interpolated from the ECLIPSE v6b inventory for 2018). In conclusion, using the CMAQ (REASv2.1) model gave insufficient BC and CO emissions from PHL, with BC emissions more inadequate than CO emissions (Sect. 3.1.2), but the BC/CO ratio calculated using the REASv2.1 was appropriate. Scaling the activity data upward in the REASv2.1 might help correct the difference between the predicted and observed concentrations by a factor of 1.2 for BC and CO, based on the observation/model ratios for E-AS-03 S2 (Figure 6c and 4d) and the constraint of a good relationship between the modeled and observed BC/CO ratios for E-AS-03 S2 (Figure 10d). However, the emission ratios for other bottom-up inventories, including the newer REASv3.2 require 10%–30% decreases in the BC/CO ratios and 25%–50% decreases in the CO/CO$_2$ ratios to match the observed ratios (Figure 11a and 9b). The underestimation of the REASv2.1 CO emission rates and the overestimation of the CO/CO$_2$ ratios indicated that the CO$_2$ emission rates for PHL in the inventory might need to be higher.

### 3.2.3 JPN case

The observed BC/CO, CO/CO$_2$, and BC/CO$_2$ ratios for the JPN case were 4.5±0.4 ng m$^{-3}$ ppb$^{-1}$ ($R = 0.71$), 13.0±0.6 ppb ppm$^{-1}$ ($R = 0.90$), and 60±6 ng m$^{-3}$ ppm$^{-1}$ ($R = 0.67$), respectively. The CMAQ model did not capture the observed BC maximum at an altitude of ~2.5 km, and the simulated BC/CO ratio was only 1.8±0.06 ng m$^{-3}$ ppb$^{-1}$ ($R = 0.95$). An unacceptably low x-axis intercept (34 ppb, which must correspond to CO baseline concentration) suggested that the simulated slope cannot be interpreted as an emission ratio but rather indicates mixing of two sources. The measured ratio was slightly lower than the 2018 BC/CO ratios for JPN determined from other ground-based observations (6.8±0.2 ng m$^{-3}$ ppb$^{-1}$ (2010–2016) (Choi et al., 2020), 5.9±3.4 ng m$^{-3}$ ppb$^{-1}$ (2009–2015) (Kanaya et al., 2016) (yellow diamonds in Figure 11d), and 5.7±0.9 ng m$^{-3}$ ppb$^{-1}$ (2003–2005)) and aircraft-based observations (6.3±0.5 ng m$^{-3}$ ppb$^{-1}$ (2003) (Kondo et al., 2006)). As shown in Figure 11d, the observed BC/CO ratios in this study were close to the ratios determined from the HTAPv2.2z (JEI-DB) emission inventory (5.15 ng m$^{-3}$ ppb$^{-1}$, shown as an open black square), the ECLIPSEv6b inventory (dark grey circles), and the EDGARv6.1 inventory (dark blue circles with a line), in which updated transport activity and shipping emission data were used (Klimont et al., 2017; https://edgar.jrc.ec.europa.eu/index.php/dataset_ap61, last accessed 22 February 2024). However, emission inventories such as CEDS (CMIP6) (Hoesly et al., 2018) (blue line with circles) and CEDS v_2021_02_05 (O'Rourke et al., 2021) (orange line with circles) clearly overestimated the observed BC/CO ratios (Figure 11d). The very high observed CO/CO$_2$ ratio (13.0±0.6 ppb ppm$^{-1}$), which was much higher than the ratio recorded at Yokosuka (JPN) in 2022 (~5.8 ppb ppm$^{-1}$), indicated unusual sources such as engineering vessels (Zhang et al., 2016; Wu et al., 2022). Ratios calculated from





other 15 yr back observations of 14–18 ppb ppm$^{-1}$ (Takegawa et al., 2004), 11.2±0.4 ppb ppm$^{-1}$, and 12.6±0.7 ppb ppm$^{-1}$ in

2003–2005 (Kondo et al., 2006) were two–three times higher than the ratio for Yokosuka. The narrow range of measured $CO_2$ concentrations (411–413 ppm) and the shallow mixing depth profile around the measuring points (determined from the HYSPLIT results, data not shown) might indicate mixing of multiple background air masses through convective transport (Yokelson et al., 2013) or the effects of long-range transport (over 5 d) from North China. We concluded because of the uncertain results, including the high $CO/CO_2$ ratios, narrow range of $CO_2$ concentration increases, and shallow mixing depths

during transport, that the air mass does not provide enough emission information for a typical Japanese air mass despite the BC/CO ratio being similar to other observed ratios and ratios calculated from bottom-up emission inventories.

### 3.2.4 CHN case

The observed BC/CO, $CO/CO_2$, and $BC/CO_2$ ratios for the Chinese air masses were 3.5±0.1 ng m$^{-3}$ ppb$^{-1}$ ($R = 0.87$), 21.1±0.4 ppb ppm$^{-1}$ ($R = 0.85$), and 77±2 ng m$^{-3}$ ppb$^{-1}$ ($R = 0.79$), respectively. These slopes did not change markedly when a slight

rain fell (by 1.5%–2.3% between full data with APT3 max=1.3 mm vs APT3=0 mm, Fig. S6 in the supplement), which could give useful information about anthropogenic emissions from CHN ($p$-value < 0.0001).

The CMAQ model gave a high BC/CO ratio of 10.5±0.1 ng m$^{-3}$ ppb$^{-1}$ ($R = 0.99$) because BC emissions were overestimated and CO emissions were underestimated. The observed BC/CO ratio was similar to the lower end of the range of ratios found for other ground-based observations made in South Korea and JPN and for ship-based observations in the East China Sea for

Chinese air masses ((6.2–7.9)±0.7 ng m$^{-3}$ ppb$^{-1}$ (Choi et al., 2020), 3.8–7.0 ng m$^{-3}$ ppb$^{-1}$ (Kanaya et al., 2020), 1.8–3.5 ng m$^{-3}$ ppb$^{-1}$ (Guo et al., 2017), and 3–6 ng m$^{-3}$ ppb$^{-1}$ (Zhu et al., 2019)). The last BC concentrations were measured using a different technique (an aethalometer), which required a correction factor for the non-BC/BC mass ratio (Zhu et al., 2019).

The observed BC/CO, $CO/CO_2$, and $BC/CO_2$ ratios for Chinese air masses (red triangles) and the ranges calculated from separate analyses of the seven sub-segments (Sect. 3.1.4, red boxes) are shown in Figure 11g–11i. Other observations and

ratios from often-used bottom-up emission inventories are also shown. The observed BC/CO ratios were less than a third of the emission ratio in the HTAPv2.2z emission inventory (12 ng m$^{-3}$ ppb$^{-1}$, shown as a black square) and only half of the ratios found for other emission inventories (8.4–16.4 ng m$^{-3}$ ppb$^{-1}$). The CEDS v_2021_02_05 inventory (O'Rourke et al., 2021), which has lower BC emissions for CHN than the earlier version (CEDS (CMIP6)), gave the lowest ratio of all the emission inventories for 2018 (10.1 ng m$^{-3}$ ppb$^{-1}$). However, this was still higher than the ratios given by the EMeRGe-Asia data and

other measurements. This introduced positive bias to the BC/CO emission ratio for CHN in the bottom-up inventories, consistent with the results of previous studies in which REASv2.1 emissions were analysed (Choi et al., 2020).

The observed $CO/CO_2$ ratio (21.6 ppb ppm$^{-1}$) was similar to the $CO/CO_2$ ratio given by the CEDS v_2021_02_05 data (23.2 ppb ppm$^{-1}$) but lower than the ratio of 28.2 ppb ppm$^{-1}$ measured during the KORUS-AQ campaign in 2016 (Tang et al., 2018). The estimated $CO/CO_2$ ratio range (15.5–23.2 ppb ppm$^{-1}$) for the seven sub-segments agreed well with the ratios calculated

from the REASv3.2 and ECLIPSEv6b inventories for 2018 (Figure 11h). This indicated that a reliable $CO/CO_2$ emission ratio



was determined using the bottom-up emission inventories for CHN. $CO_2$ emissions in the bottom-up inventories had lower uncertainties than CO and BC emissions (Zhao et al., 2013), so the similar $CO/CO_2$ ratios indicated that CO emissions estimated from the 2018 emission inventories were reasonable. The observed $CO/CO_2$ ratio was reduced from past values in the range 28–60 ppb ppm$^{-1}$ calculated from airborne measurements made ~20 yr ago (Takegawa et al., 2004; Suntharalingam

et al., 2004), confirming dramatically improved combustion efficiencies in CHN (Wang et al., 2010; Tang et al., 2018). Additional analysis of the observed $BC/CO_2$ ratio confirmed that BC emissions in CHN are overestimated in the bottom-up emission inventories because the $BC/CO_2$ ratio of 77 ng m$^{-3}$ ppm$^{-1}$ is less than half of the $BC/CO_2$ ratios calculated from other emission inventories (190–234 ng m$^{-3}$ ppm$^{-1}$) (Figure 11i).

The residential sector is the dominant source of Chinese BC and CO emissions in the HTAPv2.2z emission inventory (59%

and 54%, respectively). Other emission inventories, such as ECLIPSEv6b, also overestimate Chinese BC emissions, so detailed information about the use of different fuel types and different combustion techniques may indicate which part needs revising. The overestimated BC emissions may be related to the high emission factors for BC assumed for raw coal used in traditional cooking and heating stoves (0.06–5.76 g kg$^{-1}$), which had the first and fourth highest activity levels (51% and 5.4%, respectively), and contributed the first and second highest to BC emissions (80% and 16.6%, respectively) among various

types of residential activities (Klimont, personal communications, 2021). Emission factors have been found to be the primary factors affecting overall emission uncertainty; the reported factors are greatly dependent on the fuel quality and combustion conditions (Li et al., 2017; Zhao et al., 2013). Medium-volatile bituminous coal used in raw chunks has been found to have higher emission factors (>10.1 g kg$^{-1}$) than other types of fuel (Chen et al., 2006, 2009; Sun et al., 2017; Zhang et al., 2008; Zhi et al., 2008). This type of emission should have been progressively replaced with better techniques (e.g., honeycomb-coal-

briquette) to meet the Chinese National Air Quality Action Plan 2013 and Clean Coal Technology Policies Stage 2015 (Chen et al., 2019; Zhang et al., 2022). This portion of BC emissions was estimated to have decreased to 16.7 Gg in 2020 (Chen et al., 2009) even though 401 Gg were assigned in the ECLIPSEv6b inventory.

Emissions of BC might be more heterogeneous than emissions of CO and $CO_2$ both spatially and temporally. As shown in Figs. S3 and S4, the NS-CEC air masses had BC/CO ratios < 2.9 ng m$^{-3}$ ppb$^{-1}$ (2.4–2.7 ng m$^{-3}$ ppb$^{-1}$) but the S-CEC air masses

had BC/CO ratios as high as 3.5 ng m$^{-3}$ ppb$^{-1}$ (3.6–4.7 ng m$^{-3}$ ppb$^{-1}$). This might be related to changes in emissions in South and North China between 2010 and 2018 resulting in the BC/CO ratio decreasing more in N-CEC than S-CEC (Kanaya et al., 2020). The airborne BC/CO ratios for S-CEC air masses were similar to the ratios calculated from observations of the same transport episode at Fukue Island (3.5 ng m$^{-3}$ ppb$^{-1}$; Table S4), indicating that the BC/CO ratio did not change markedly during transport. These ratios were markedly lower than the spring mean ratios for N-CEC air masses (4.4 ng m$^{-3}$ ppb$^{-1}$) and S-CEC

air masses (5.2 ng m$^{-3}$ ppb$^{-1}$), recorded at Fukue Island. BC/CO emission ratios calculated from airborne measurements may not uniformly represent all regions of CHN, but we found for two flights that the observed BC/CO ratio was constant from the ground to the aircraft altitude. Future airborne observations with larger footprints will allow more accurate estimates of countrywide emissions to be made.



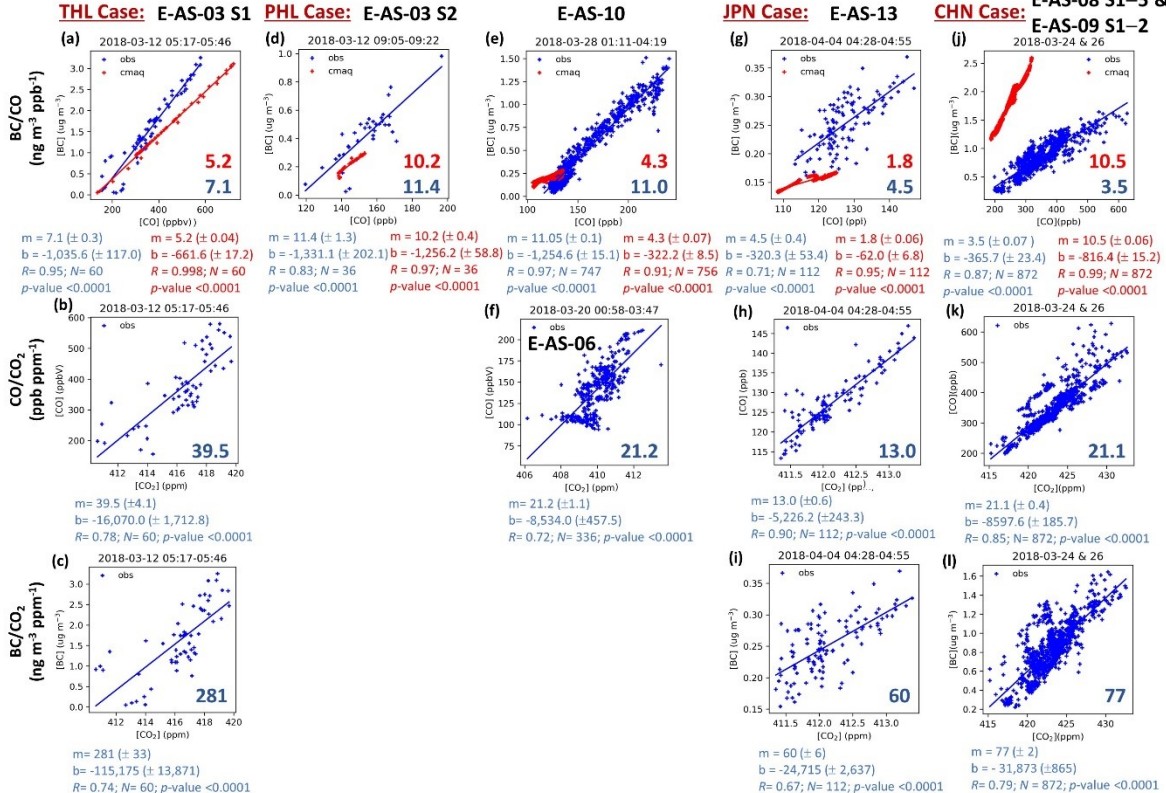


**Figure 10. Residual BC/CO, CO/CO₂, and BC/CO₂ ratios, estimated as slopes from regression lines, for the (a–c) Thailand (THL) case, (d–f) Philippines (PHL) case, (g–i) Japan (JPN) case, and (j–l) China (CHN) case. Blue dots represent airborne data from the Effect of Megacities on the Transport and Transformation of Pollutants at Regional and Global Scales (EMeRGe) campaign with linear fitted lines (blue lines) for BC and CO, and similarly red dots and lines represent data simulated using the Community**
**Multiscale Air Quality (CMAQ) model. Blue and red text show the slopes (m), intercepts (b), correlation coefficients (R), data points (N), and p-values for the lines fitted to the observed and simulated data, respectively.**



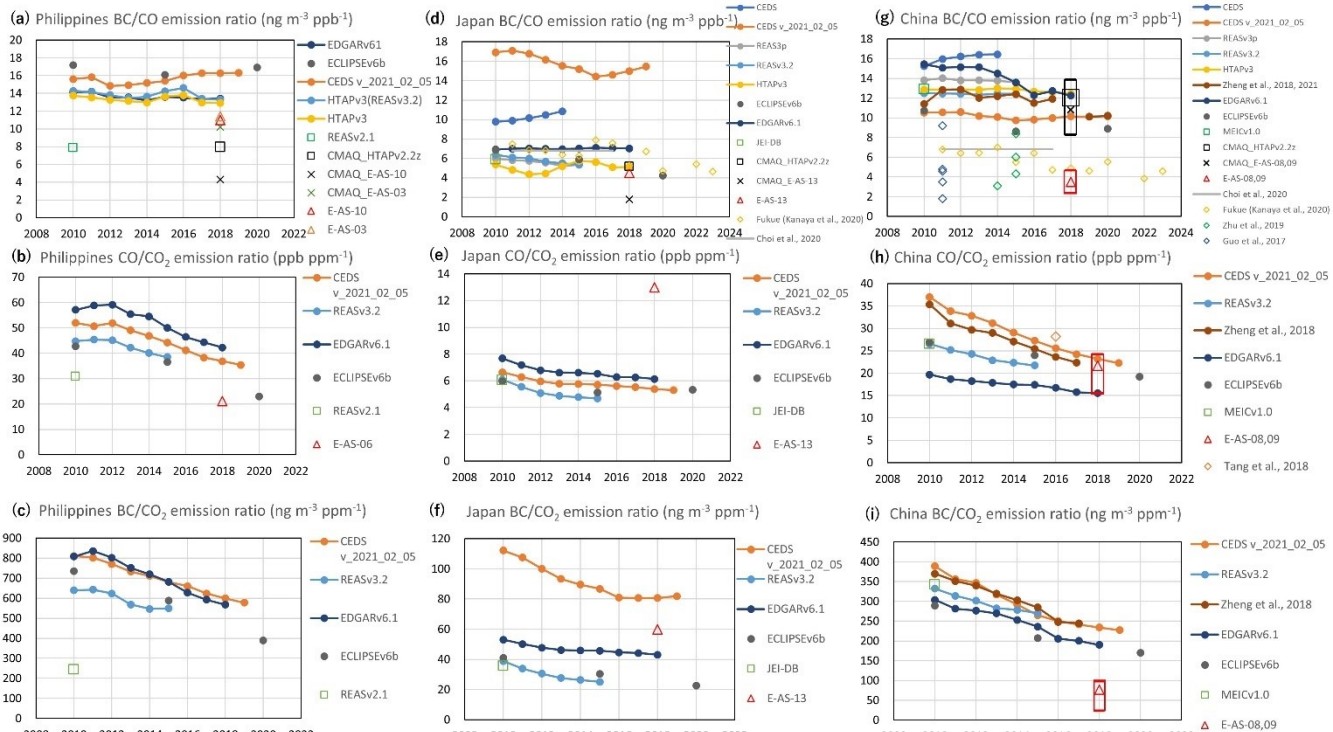

**Figure 11. Residual BC/CO, CO/CO₂, and BC/CO₂ ratios for the (a–c) Philippines (PHL) case, (d–f) Japan (JPN) case, and (g–i) China (CHN) case measured by aircraft-based instruments (open red triangles) and simulated using the Community Multiscale Air Quality (CMAQ) model (crosses). In (g–i), the boxes indicate the ranges for the measured (red) and simulated (black) ratios. Residual ratios for other observations (open diamonds and grey lines), mosaic Asian anthropogenic emission inventory (MIX; green squares), Hemispheric Transport of Air Pollution (HTAP) v2.2z emission inventory (black squares), and other emission inventories (solid markers with or without lines) are also shown.**

## 4 Conclusions

An integrated analysis of combustion-related SLCF emission in East Asia was performed. Airborne BC, CO, and $CO_2$ measurements, simulations performed using the Weather Research and Forecasting/CMAQ model (v5.0.2), and various emission inventories were used. Four different pollution case studies from March and April 2018 were investigated. These were emissions through combustion near the THL and urban pollution from PHL, JPN, and CHN. We drew the following conclusions from the study.

a)  Using the GFEDv4.1s inventory for fire emissions in the THL case gave accurate BC and CO emissions. Analysing the residual ratios of co-emitted pollutants gave information about emission ratios. The BC/CO concentration ratio was 7.1±0.3 ng m⁻³ ppb⁻¹ for emissions through flaming- dominant combustion over the Indochina Peninsula.

b)  Simulations of the PHL air masses indicated that emissions in 2010 in the REASv2.1 underestimated the BC and CO concentrations in the Manila region. The BC/CO concentration ratio for urban emissions from Manila was 11.4±1.3 ng m⁻³



ppb$^{-1}$. The REASv2.1 gave a good range of BC/CO ratios for the Manila area (10.2±0.4 ng m$^{-3}$ ppb$^{-1}$). We concluded that the underestimation in REASv2.1 should be likely with the activity information.

c) The air mass for the JPN case had a high CO/CO$_2$ ratio that might not be typical of a Japanese air mass despite the BC/CO ratio being close to ratios found from other observations and bottom-up emission inventories.

d) For the CHN air masses, aircraft observations confirmed that the BC concentration was overestimated (+1.62 µg m$^{-3}$)
and the CO concentration was underestimated (−400 ppbv) by the CMAQ model using HTAPv2.2z emission inventory, supporting the conclusion drawn from ground-based observations (Kanaya et al., 2020); The BC/CO ratio for Chinese air masses found in Kanaya et al. (2020) was confirmed for the first time using the data acquired by aircraft-based instruments in this study (3.5±0.1 ng m$^{-3}$ ppb$^{-1}$). This BC/CO concentration ratio agreed with ratios found from other ground-based and ship-based observations but was markedly lower than ratios found using the CMAQ model (10.5±0.1 ng m$^{-3}$ ppb$^{-1}$) and other
emission inventories. This implied that BC emissions were overestimated and CO emissions were underestimated in the HTAPv2.2z emission inventory and other emission inventories. This might have been caused by a rapid decrease in BC emissions through coal combustions and the use of old-fashioned heating stoves in CHN in the decade to 2018.

e) A combined model–observation approach was used to estimate BC, CO, and CO$_2$ emission rates for CHN. The best estimates used the spring mean residual BC/CO ratio 4.9±0.1 ng m$^{-3}$ ppb$^{-1}$ and the aircraft-measured CO/CO$_2$ ratio 21.1±0.4
ppb ppm$^{-1}$. The emission estimates for CHN were 0.65±0.25 (Tg BC) yr$^{-1}$, 166±65 (Tg CO) yr$^{-1}$, and (12.4±4.8)×10$^3$ (Tg CO$_2$) yr$^{-1}$. The BC and CO emission estimates were tied to CO$_2$ emissions to reduce uncertainty. The low BC emission rate implies that BC emission from CHN in the bottom-up HTAPv2.2z emission inventory needs to decrease by ~50% and CO emission needs to increase by ~20% to improve agreement with airborne measurements.

The results indicate that bottom-up emission inventories in which gaps were found need to be verified to improve air
quality simulations and climate impact mitigation strategies. The overestimated BC emissions from CHN in the emission inventories imply that the warming effect of BC is smaller than previously estimated in the CMIP6 model, as suggested by Ikeda et al. (2023).

**Data availability**

The primary data collected by HALO aircraft during the EMeRGe campaign are publicly available and can be requested from
the HALO database (https://halo-db.pa.op.dlr.de/). The data simulated using the CMAQ model can be acquired upon request to Kazuyo Yamaji (kazuyo@maritime.kobe-u.ac.jp).

**Author contributions**

P Ha conducted all the analysis and composed the text. Y Kanaya supervised the findings of this study and contributed significantly to manuscript revisions. K Yamaji owns the CMAQ model code and provides the model's inputs and results. S

Itahashi provided Comprehensive Air quality Model with eXtensions data of BC and CO concentrations, which was used to calculate the error estimate for the transport. S Chatani provided the model-ready emission inputs. T Sekiya provided the CO_TCR2 data. Y Kanaya, K Yamaji, and T Sekiya thoroughly discussed the analysis results before and during the manuscript writing. MD Andrés Hernández, JP Burrows, H Schlager, M Lichtenstern, M Poehlker, B Holanda provided the EMeRGe-Asia campaign's data used in the study. All authors have contributed to the manuscript, corrections, post-writing formatting,
and revisions in reviewing and interpreting the results presented.

**Competing interests**

At least one of the (co-)authors is a member of the editorial board of Atmospheric Chemistry and Physics.

**Acknowledgements**

We thank Z Klimont for the calculation details in the ECLIPSEv6b inventory for BC emissions used in Sect. 3.2.4. Earth
Simulator was used for the data assimilation calculation of CO_TCR2 data via the Japan Agency for Marine-Earth Science and Technology support. We thank Gareth Thomas, PhD, from Edanz (https://jp.edanz.com/ac) for editing a draft of this manuscript.

**Financial support**

This study is part of the ERTDF 2-2201 project, which has been supported by the Environmental Research and Technology
Development Fund (ERTDF) from the Ministry of the Environment, Japan (grant no. 2-2201: JPMEERF20222001). The ERTDF grant has supported the simulations using the CMAQ model no. 5-1601: JPMEERF20165001 and JSPS KAKENHI Grant Number 23K25011. The HALO deployment during EMeRGe was funded by a consortium comprising the German Research Foundation (DFG) Priority Program HALO-SPP 1294, the Institute of Atmospheric Physics of DLR, the Max Planck Society (MPG), the Helmholtz Association and the RCEC Academia Sinica in Taiwan. The University of Bremen supported
in part the EMeRGe measurements and the contributions made by MDAH and JPB.



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
