# Peer review of "Assessing BC and CO Emissions from China Using EMeRGe Aircraft Observations and WRF/CMAQ Modelling"

_EGUsphere, 2024_

## Referee Comment (RC1)

**Comments on ACP 2024-2064**

**Downward and upward revisions of Chinese emissions of black carbon and CO in bottom-up inventories are still required: an integrated analysis of WRF/CMAQ model and EMeRGe observations in East Asia in spring 2018**

**General Comments**

The manuscript "Downward and upward revisions of Chinese emissions of black carbon and CO in bottom-up inventories are still required: an integrated analysis of WRF/CMAQ model and EMeRGe observations in East Asia in spring 2018" focuses on constraining an inventory based on aircraft observations and modeling. The authors used aircraft observations of black carbon and carbon monoxide from a 2018 campaign, WRF/CMAQ forward modeling, and back-trajectory tools to estimate the mismatch between predicted and observed aerosol concentrations over East and Southeast Asia. The findings suggest a potential bias in current inventories for black carbon, carbon monoxide, and carbon dioxide over the study regions and also show that the modeled aerosol concentrations produced with this model and a wildfire emissions inventory are within reasonable bounds of observed values. As such, the findings of this work are timely and can have an impact on informing emission inventory development studies for this region. The manuscript contains detailed analyses and is well-written. However, the flow can be improved with better organization. Moreover, the extrapolation of aircraft observations of a few days to an annual inventory over a large region should be justified. Specific comments are below.

Overall, I recommend the manuscript for publication after minor analyses-based revisions but more writing- and presentation-based revisions.

**High-level comments**

1.  Consider reorganizing the manuscript for clearer flow. Current flow does not read well and may be improved by aggregating analyses based on a region, adding an Overview in the Methods, and then a bulleted list of analyses performed. Similar changes in the Results section could be useful. Also consider renaming section titles to be clearer and consistent.

2. The manuscript needs more focus on the robustness of the results when compared to aircraft observations. Do a few days of aircraft observations represent seasonal or annual magnitudes? Or could the uncertainty in the modeled concentrations be more due to the simulated meteorology and aerosol representation alone and not due to the underlying inventory?

3. While a model-observation study shows potential bias in the inventory, it does not separate those biases by source. Which sources would inventory developers need to tune up or down to match these observations? Can aircraft observations aid in identifying hotspots from a region?

4. The model-observation mismatch is attributed to uncertain emissions in the inventory. However, as the HYSPLIT back-trajectories show, the source regions could be a narrow band or can cover multiple regions. How robust are this study's findings in generalizing over the entire CHN or other regions?

5. Consider consolidating Figures 2-5 into one figure and focus on showing how the model performs in each of the cases. For example, a figure could focus on just the identification of case names based on observed back-trajectories, and another figure could focus on the flight observations and modeled concentrations. Show the relevant flight paths in the figures with back trajectories.

6. Aircraft observations have their own merits and demerits. For example, while they aid in isolating non-local sources, comparing them with modeled concentrations is difficult due to higher model uncertainties at those distances from the source region. At aircraft distances, model uncertainties may be higher than emission inventory uncertainties, for example, due to modeled rapid deposition offsetting higher emissions. How robust are aircraft observations at isolating uncertainties in emission inventories?

**Specific Comments**

**Title:** It can be made tighter; something along the lines of "Assessing uncertainty in emission estimates from China using EMeRGe aircraft observations and models".

**L22:** See point 6 in high-level comments. Justify this in the Introduction.

**L30:** "The results suggested that downward and upward revisions of Chinese emissions of BC (−50%) and CO (+20%), respectively, are required in HTAPv2.2z emission inventory." Also mention the range in other inventories such as CEDS (Hoesly et al., 2018) or the IIASA GAINS.

**L34:** Mention the lifetimes of SLCFs.

**L50:** "Our understanding of the responses of SLCF emissions to the establishment of techniques that decrease emissions in the last two decades in fast-growing Asian economies is insufficient (Chen and Chen, 2019; Kanaya et al., 2020; Ikeda et al., 2023; Zhang et al., 2022)." This point was not clear. Emissions decrease as we use more abatement (assuming activity remains the same), and the same is true everywhere.

**L52:** "Biomass-burning habits in Southeast Asia has become the main contributor of carbon emissions from forest fires in spring (Reid et al., 2013; Heald et al.,2003; Palmer et al., 2006; Johnston et al., 2012)." Consider rewording this to not write it as a 'habit'. People use biomass as a source of residential or heating energy in those regions and while it is a habit, it is more of a need.

**L80:** Good point. Agree!

**L83:** "Emission inventories therefore need to be tested using independent observational data." I understand this focuses on the EMeRGe dataset, but any mention of ground-based long-term monitoring and its usefulness would also be good to include in the Introduction, and then mentioning how aircraft observations can help provide another perspective, will be great to add.

**L132-137:** I appreciate the inclusion of underlying inventories in HTAP.

**L164:** Why is 1200hrs (or 5 days) a suitable time range for back-trajectory analyses in this work?

**L189:** Describing the common BC/CO, CO/CO2, and BC/CO2 ratios in different combustion sources would aid the reader in understanding that whenever there is a higher BC/CO ratio, it signals a source X, and a lower CO/CO2 ratio signals source Y.

**L215:** Shouldn't this correction factor be applied to only those grids in CHN where the emissions are coming from and were measured in the campaign? Or to the whole of CHN?

**Equations 4, 5, and 6:** Should this correction factor based on BC/CO and other ratios be applied to just the emission source with a known BC/CO (and other ratios)? For example, if the aircraft observed a high BC/CO ratio, it signals those emissions from source X dominated the concentrations on that day. Now, the modeled concentrations show a lower BC/CO ratio, indicating either the contribution from source X in the inventory is low, or that the contribution from other sources (with a lower BC/CO ratio at emissions) is higher. How do you differentiate between these two offsetting effects?

**L225:** This paragraph should be in Methods.

**L260:** "On the other hand, the IFS-CAMS simulation predicted the maximum CO concentration well in Deroubaix et al. (2024a), possibly because anthropogenic emissions in the IFS-CAMS simulation were taken from the CAMS-GLOB-ANTv4.2 emission inventory (Granier et al., 2019)." This brings up a good point and something that should be discussed in the end and generally --- how does the choice of inventory affect your findings?

**L264:** A general comment --- what does the observed/modeled ratio refer to? Is it the average observations and average modeled concentrations for the whole flight path? If so, the values closer to the source region better represent emissions uncertainty due to lesser influence by transport processes such as coagulation and deposition. Consider adding any discussion on that as well.

**L282:** This raises a good point about the temporal resolution in the inventory. Inventories probably do not capture diurnal or hourly emission patterns and thus the bias in aircraft observations at two time-stamps against modeled concentrations may be due to the lack of such temporal resolution in inventories. Consider mentioning that in perspective of the model-observation differences in this and other sections or in general.

**Table 2:** How is the R calculated here? Why is it so low in some cases? Consider adding Spearman's coefficient since the model and observed trends were similar.

**L339:** Even if there is no rainfall, higher moisture can lead to rapid aerosol growth from condensation and thus a faster deposition, especially farther from source regions when BC gets activated due to sulfate reactions.

**L377:** Similar to the comment above on Equations 4, 5, and 6 --- should not this correction be based on source-specific BC/CO ratios? This can also explain some of the differences observed in Figures 7a and 7b.

**L460-465:** While this is a great finding, consider including some process-specific discussion --- which specific source needs to be tuned up or down in inventories? Do the observed BC/CO/CO2 ratios help in differentiating source signals?

**Figure 6:** Any explanation on why the model does not produce as much variability (concentration range) as observations will be good to add.

**Figure 8:** How did you calculate the uncertainty in red and green boxes?

**Sec. 3.2:** This section touches upon some of the points I raise above but a richer discussion in identifying the sources will be good to add. Consider using information from this section to inform inventory updates in the previous section. This does not need to be big changes but a short circling back would be useful.

**Sec. 4:** Consider adding recommendations on how inventory-developers can directly utilize such campaigns' information to tune up or down in this section.

**L653:** Add discussion on whether there is any chance or reason why the BC/CO ratios in the model and observations are similar due to completely different reasons, such as rapid BC deposition in the model offset by higher BC emissions?

---

## Author Comment (AC1)

Authors' replies on Referee#1's Comments on ACP 2024-2064

"Downward and upward revisions of Chinese emissions of black carbon and CO in bottom-up inventories are still required: an integrated analysis of WRF/CMAQ model and EMeRGe observations in East Asia in spring 2018"

**General Comments**

**AC:** We thank for your time and efforts in reviewing our manuscript. We are grateful for your general comments on recommending it to be published after revisions. We also acknowledge the benefit of writing- and presentation-based revisions for our manuscript. We have thoroughly revised the manuscript, noting the changes and indicating the line numbers in the author comments for each referee comment. Minor textual adjustments and reconstructed text are not highlighted.

**High-level comments**

RC: 1. Consider reorganizing the manuscript for clearer flow. Current flow does not read well and may be improved by aggregating analyses based on a region, adding an Overview in the Methods, and then a bulleted list of analyses performed. Similar changes in the Results section could be useful. Also considering renaming section titles to be clearer and consistent.

AC: We now have the overview or summary in the Methods **(L100-104)** and Results **(L284-287)** by moving the relevant text to this section, and the Conclusion **(L644-646)**.

We substantially reorganized the manuscript by aggregating analyses based on regions. Section 3 (Results) now includes 2 subsections: 3.1 for three polluted cases (THL, PHL, JPN) including comparisons of concentrations and residual ratios; 3.2 for CHN case including comparisons of concentrations, residual ratios, and emission estimates, with a separate part for uncertainty assessments. **Section titles** were also renamed.

RC: 2. The manuscript needs more focus on the robustness of the results when compared to aircraft observations. Do a few days of aircraft observations represent seasonal or annual magnitudes?

**AC (L461-464, 606-609)**: The analyzed plume originating from a major pollution center in China was one of the most significant air pollution transport events recorded during EMeRGe and by our long-term ground-based observations at Fukue Island in western Japan over the past 10 years.

The event was regarded as representative with respect to its transportation route and negligible deposition loss **(L461-464)**. The annual and nationwide emission estimates appear to be extrapolations based on limited aircraft observations. However, this extrapolation is supported by our previous analysis, which used long-term ground-based observations (Kanaya et al., 2020). This point has been clarified in the revised text (**L606-609**). Note that the uncertainty associated with representativeness has been discussed in the original manuscript ("We estimated (3) representation errors…", L582-585).

RC: 2. Or could the uncertainty in the modeled concentrations be more due to the simulated meteorology and aerosol representation alone and not due to the underlying inventory?

AC (**L588-590**): The difference between the model and observations was investigated using another model (CAMx), which employed different meteorology and aerosol schemes but used emission inventory common to the CMAQ-based simulations. The analysis with the CAMx resulted in a correction factor of 0.54, similar to 0.48 from CMAQ. This suggests that the discrepancy is primarily due to the underlying emission inventory, and not from the meteorology or aerosol scheme. We added some more explanation in the part for model-specific uncertainty assessment (**Sect. 3.2.4**).

RC: 3. While a model-observation study shows potential bias in the inventory, it does not separate those biases by source. Which sources would inventory developers need to tune up or down to match these observations? Can aircraft observations aid in identifying hotspots from a region?

**AC (L606, 523-526):** It is difficult to point to sectors from our observation only. Nonetheless, the analysis of the BC/CO enhancement ratio implies that the BC and CO emissions in the model need to be reduced and increased, respectively, providing hints that sectors with large contributions or very high BC/CO emission ratio have to be tuned up. They included coal-fuel stoves. A thorough review of individual sources also helps assessing missing updates that should reflect policy changes or new experimental data. We added discussions in **Sect. 3.2.2 and 3.2.4**, supported by our previous analysis based on ground-based observations (Kanaya et al., 2020, 2021).

RC: 4. The model-observation mismatch is attributed to uncertain emissions in the inventory. However, as the HYSPLIT back-trajectories show, the source regions could be a narrow band or can cover multiple regions. How robust are this study's findings in generalising over the entire CHN or other regions?

AC (**L606-610**): We agree that the footprint from two flights covers the key emission regions but is not very wide, even if it is simulated taking atmospheric diffusion into account across the central transport axis, which was not included in the original HYSPLIT trajectories. As stated earlier, the annual-based and country-wide emission estimation is an apparent extrapolation from the analysis of the limited number of events, but the idea has been supported by our previous analysis based on the long-term ground-based observations (Kanaya et al., 2020). These points have been clarified in the revised text. Note that the associated uncertainty arising from representativeness has been discussed in the original manuscript (L582-585).

RC: 5. Consider consolidating Figures 2-5 into one figure and focus on showing how the model performs in each of the cases. For example, a figure could focus on just the identification of case names based on observed back-trajectories, and another figure could focus on the flight observations and modeled concentrations. Show the relevant flight paths in the figures with back trajectories.

**AC:** Thank you for the suggestions. We have updated the **figures** and all figure cross-references accordingly.

RC: 6. Aircraft observations have their own merits and demerits. For example, while they aid in isolating non-local sources, comparing them with modeled concentrations is difficult due to higher model uncertainties at those distances from the source region. At aircraft distances, model uncertainties may be higher than emission inventory uncertainties, for example, due to modeled rapid deposition offsetting higher emissions. How robust are aircraft observations at isolating uncertainties in emission inventories?

AC (**L58-59, 206-218, 588-590**): Model uncertainties in the aircraft observation range would be indeed non-negligible. Nonetheless, we demonstrate that the transport/meteorology error is small, by showing that a similar overestimation, occurred with a different model system (CAMx), using the same emission inventory (**L58-59, 588-590**). Note that we have attributed the transport error to be 26% in the original manuscript (L585-588).

The BC/CO emission ratio is free from such transport error and provides an unequivocal indication that the emission used must not be wrong. Consistency with the ratio from ground-based data on Fukue Island ($\approx 3.5$ ng m$^{-3}$/ppb) corroborates our conclusion (mentioned in the original manuscript at L517-519).

Deposition during transport is excluded from the captured air mass, given that the mean APT 72 h is only $0.1 \pm 0.3$ mm, indicating negligible precipitation influence. Previous studies have shown that a low APT 72 h value (e.g., < 1 mm) for the air mass with typical 40 hours transport time indicates negligible wet deposition, which does not significantly influence emission-specific characterization (Kanaya et al., 2016, 2020) (**L206-218**). In our study, the transport time of the CHN air masses to the aircraft was ~40 – 120 hours (**L201**).

We have added some text for clarification in **Introduction** part (**L58-59**) together with explanatory text to Section 2.3 and 3.2.4, addressing several related comments.

**Specific Comments**

RC: Title: It can be made tighter; something along the lines of "Assessing uncertainty in emission estimates from China using EMeRGe aircraft observations and models".

AC: We made change to the **title** as "Assessing BC and CO Emissions from China Using EMeRGe Aircraft Observations and WRF/CMAQ Modeling"

RC: L22: See point 6 in high-level comments. Justify this in the Introduction.

AC (**L58-59, 206-218, 588-590**): We added the discussion to the Introduction and relevant parts (Sect. 2.3 and 3.2.4).

RC: L30: "The results suggested that downward and upward revisions of Chinese emissions of BC (−50%) and CO (+20%), respectively, are required in HTAPv2.2z emission inventory." Also mention the range in other inventories such as CEDS (Hoesly et al., 2018) or the IIASA GAINS.

AC (**L30-34**): We added in the **abstract**.

RC: L34: Mention the lifetimes of SLCFs.

AC (**L36-37**): We added the lifetimes of SLCFs as requested in the **introduction**.

RC: L50: "Our understanding of the responses of SLCF emissions to the establishment of techniques that decrease emissions in the last two decades in fast-growing Asian economies is insufficient (Chen and Chen, 2019; Kanaya et al., 2020; Ikeda et al., 2023; Zhang et al., 2022)." This point was not clear. Emissions decrease as we use more abatement (assuming activity remains the same), and the same is true everywhere.

AC (**L52-53**): It was changed to "The changes in **national** SLCF emissions due **to the potentially uneven adoption of emission reduction techniques** across fast-growing Asian economies over the past two decades are not well understood (Chen and Chen, 2019; Kanaya et al., 2020; Ikeda et al., 2023; Zhang et al., 2022).”

RC: L52: “Biomass-burning habits in Southeast Asia has become the main contributor of carbon emissions from forest fires in spring (Reid et al., 2013; Heald et al.,2003; Palmer et al., 2006;
Johnston et al., 2012).” Consider rewording this to not write it as a 'habit'. People use biomass as a source of residential or heating energy in those regions and while it is a habit, it is more of a need.

AC (**L56):** It was changed to “Moreover, biomass-burning in Southeast Asia has become the
primary source of carbon emissions from forest fires in spring (Reid et al., 2013; Heald et al., 2003; Palmer et al., 2006; Johnston et al., 2012), **driven by the regional reliance on biomass for residential and agricultural purposes**.”

RC: L80: Good point. Agree!
AC: We appreciate your support on this point.

RC: L83: “Emission inventories therefore need to be tested using independent observational data.” I understand this focuses on the EMeRGe dataset, but any mention of ground-based long-term
monitoring and its usefulness would also be good to include in the Introduction, and then mentioning how aircraft observations can help provide another perspective will be great to add.

AC (**L58-59, 206-218, 588-590**): We believe the addition in response to the comment on L22 adequately addresses this point.
RC: L132-137: I appreciate the inclusion of underlying inventories in HTAP.

AC (**L155-158**): We appreciate your support on this point.

RC: L164: Why is 1200hrs (or 5 days) a suitable time range for back-trajectory analyses in this work?

AC (**L195-200):** A 120-hour (5-day) period was selected for the back-trajectory analyses in long-range transport studies, to account for the time it takes for air masses to travel from the time of
emissions to the time of detection by the aircraft (traveling time for the studied events was estimated to be 40–120 hours for the CHN case). This approach is consistent with previous studies (Choi et al., 2020; Kanaya et al., 2016; Kanaya et al., 2020; Miyakawa et al., 2017; Zhu et al., 2019).

RC: L189: Describing the common BC/CO, CO/CO2, and BC/CO2 ratios in different combustion sources would aid the reader in understanding that whenever there is a higher BC/CO ratio, it signals a source X, and a lower CO/CO2 ratio signals source Y.

AC (**L234-239**): We added the description.

RC: L215: Shouldn't this correction factor be applied to only those grids in CHN where the emissions are coming from and were measured in the campaign? Or to the whole of CHN?

AC **(L268-270):** Considering that emission structures are almost uniform over the country, as
evidenced from our longer-termed ground-based observations receiving various air masses, we concluded that it is reasonable to apply the correction factor to the entire region of China.

RC: Equations 4, 5, and 6: Should this correction factor based on BC/CO and other ratios be applied to just the emission source with a known BC/CO (and other ratios)? For example, if the
aircraft observed a high BC/CO ratio, it signals those emissions from source X dominated the concentrations on that day. Now, the modeled concentrations show a lower BC/CO ratio, indicating either the contribution from source X in the inventory is low, or that the contribution from other sources (with a lower BC/CO ratio at emissions) is higher. How do you differentiate between these two offsetting effects?

AC **(L277-282):** Differentiating between these two offsetting effects is difficult because Chinese emissions are well-mixed before reaching the aircraft, as sector-specific sources exhibit partial spatial and temporal overlap in emission inventories. To estimate total national emissions, we applied averaged ratios and correction factors from captured air masses and the model-prescribed
national emission amounts, ensuring a more representative result by mitigating local transport and source biases. To minimize BC/CO ratio discrepancies in aircraft measurements, we used the spring-averaged value from Fukue Island. Further explanation is given in the end of **Sect. 2.5**.

RC: L225: This paragraph should be in Methods.
AC **(L206-218):** We moved a part of it (introduce the precipitation and APT properties of the selected cases) to the end of Sect. 2.3.

RC: L260: "On the other hand, the IFS-CAMS simulation predicted the maximum CO concentration well in Deroubaix et al. (2024a), possibly because anthropogenic emissions in the IFS-CAMS simulation were taken from the CAMS-GLOB-ANTv4.2 emission inventory (Granier et al., 2019)." This brings up a good point and something that should be discussed in the end and generally --- how does the choice of inventory affect your findings?

**AC (L408-409 and 610-612):** The regional emission distribution pattern is largely consistent across inventories, with a nearly linear concentration-emission response (Ikeda et al., 2022). Therefore, the choice of emission inventory does not impact the findings of this study (**Sect. 3.2.4, L610-612**). However, selecting representative air masses is key to accurately analyzing national emission ratios, while examining co-emitted species concentrations and meteorological parameters can inform whether an air mass is sufficiently indicative of the region (**Sect. 3.1.3, L408-409**).

We added relevant discussions to the final part of the Chinese emission uncertainty (**Sect. 3.2.4**) and the JPN emission ratios (**Sect. 3.1.3**).

RC: L264: A general comment --- what does the observed/modeled ratio refer to? Is it the average observations and average modeled concentrations for the whole flight path? If so, the values closer to the source region better represent emissions uncertainty due to lesser influence by transport processes such as coagulation and deposition. Consider adding any discussion on that as well.

AC (**L151-152, 254-256**): the observed/modeled ratios were calculated using Eqs. 1 and 2. Values closer to the source region represent best emissions uncertainty due to minimal transport influence and deposition. Values far from source are affected by deposition. While coagulation alters particle size distribution, it does not remove BC mass from the air mass (**L151-152**). Wet and dry deposition for this range of transport speeds have been analyzed and found to be negligible in the case where the APT in 72 hours traveled from the source is smaller than 1 mm (Kanaya et al., 2016, 2020). Thus, we assumed linear responses between emissions and concentrations for both BC and ΔCO, which has been verified by Ikeda et al. (2022). This is discussed in **Sect. 2.5**, including an additional explanation in **L254-256** and a small mention about coagulation in **Sect. 2.2 (L151-152)**.

RC: L282: This raises a good point about the temporal resolution in the inventory. Inventories probably do not capture diurnal or hourly emission patterns and thus the bias in aircraft observations at two time-stamps against modeled concentrations may be due to the lack of such temporal resolution in inventories. Consider mentioning that in perspective of the model-observation differences in this and other sections or in general.

AC (**L340, 366-367**): Emission inventories capture typical hourly and diurnal emissions but not irregular events. Japan's JEI-DB includes diurnal anthropogenic emissions, allowing the 3–6 UTC data to reflect small enhancements from JPN emissions. Thus, observed BC concentrations fluctuated without a clear CO increase, likely due to mixed sources and low enhancement levels rather than the inventory temporal resolution. In contrast, CHN and PHL inventories use monthly emission data, where limited temporal resolution may contribute to discrepancies with observational concentrations (**L340**). However, concentration ratio gaps primarily stem from emission factors or pollutant activity levels, with certain sectoral activity levels requiring more detailed characterisation (**L366-367**). Note that the observed air masses have been influenced from emissions integrated over time along the transport, which must be longer than the observation duration, particularly for the CHN case. Thus the observed concentration differences in the 2–3 hours (from different locations) would be more reasonably attributed to the spatial inhomogeneity (due to temporarily averaged emissions and transport) than to the diurnal variation of emissions itself. We added some short explanation to the PHL case Sect. 3.1.2 (**L340, 366-367**).

RC: Table 2: How is the R calculated here? Why is it so low in some cases? Consider adding Spearman's coefficient since the model and observed trends were similar.

AC: In the THL and JPN cases, the Pearson correlation coefficient (R) is relatively low due to temporal discrepancies between the modeled and observed peak times, as well as the influence of mixed sources. We examined the Spearman coefficients, but they showed only minimal improvement over the Pearson correlation coefficients, so we opted not to include them.

RC: L339: Even if there is no rainfall, higher moisture can lead to rapid aerosol growth from condensation and thus a faster deposition, especially farther from source regions when BC gets activated due to sulfate reactions.

AC (**L207-215, 306-311, 322, 461-464**): Excluding data based on stricter rainfall/moisture criteria had to be avoided unless necessary, as the available data for each case is already limited. A detailed analysis of Accumulated Precipitation along Trajectories (APT) was conducted, applying APT criterion for the air masses from CHN, THL (APT 72 h < 1 mm), PHL E-AS-06 and E-AS-10 (APT 24 h = 0 mm), PHL E-AS-03 S2 (APT 10 h = 0 mm), JPN (APT 36 h = 0 mm) based on their traveling time to the aircraft (Table S6). The APT 72 h < 1 mm criterion has been previously analyzed and showed no large influence of wet deposition on the emission-concentration relationship (Kanaya et al., 2016, 2020), whereas the APT = 0 mm criterion aims to completely exclude the effect of rain. Compared to the full dataset, applying APT criterion reduced data across all cases except PHL (E-AS-03 S2) and JPN. In CHN and PHL (E-AS-06), statistical metrics showed only minor changes. In PHL, metrics for E-AS-10 changed due to substantial data reduction and biogenic flux complexity, though the residual ratios remained largely stable (**Sect. 2.3, L211-214**).

In THL, R values increased and observational concentrations decreased (**Sect. 2.3, L214**). The observed BC/CO ratio has decreased, bringing it closer to the modeled value, with regression slopes of $5.21 \pm 0.68$ ng m$^{-3}$ ppb$^{-1}$ (observation) and $5.37 \pm 0.07$ ng m$^{-3}$ ppb$^{-1}$ (model). While observed BC, CO, and $CO_2$ concentrations declined, simulated values remained unchanged, suggesting that the air mass selection, not the APT criterion, is responsible. The exclusion of long-range transport from Myanmar, northern Laos, Vietnam, and southern China over the East Ocean, while retaining local air masses from Cambodia, Thailand, and southern Laos, may enhance the model's representation of regional air masses (Figure S5e, f). $CO/CO_2$ and $BC/CO_2$ ratios dropped to 24 ppb ppm$^{-1}$ and 112 ng m$^{-3}$ ppm$^{-1}$, respectively, though correlation coefficients worsened. However, these ratios remain consistent with reference values for dominant fire types in THL (**Sect. 3.1.1, L.306-311, 322**)

For CHN case, the HYSPLIT back trajectory model indicates that specific humidity and relative humidity remained low throughout the three-day transport period. The mean specific humidity was $3.6\pm1.1$ g/kg, while relative humidity averaged $57\pm13\%$. These values suggest that high moisture levels or cloud formation were unlikely to influence the airmass during transport, under the high-pressure system (**Sect. 3.2.1, L461-464**).

We briefly added some relevant discussion in Method Sect. 2.3 (**L207-215**), Sect. 3.1.1 for THL (**L. 306-311, 322**), Sect. 3.2.1 for CHN (**L461-464**).

RC: L377: Similar to the comment above on Equations 4, 5, and 6 --- should not this correction be based on source-specific BC/CO ratios? This can also explain some of the differences observed in Figures 7a and 7b.

AC (**L277-278**): We explained above that this study aims to estimate national total emissions to facilitate comparisons with other references. Chinese emissions are well-mixed before reaching the aircraft or Fukue Island, as sector-specific sources (e.g., power plant, transport, domestic, industries) exhibit largely spatial and temporal overlap in emission inventories. So, the explanation in Sect. 2.5 still applies for this issue.

RC: L460-465: While this is a great finding, consider including some process-specific discussion --- which specific source needs to be tuned up or down in inventories? Do the observed BC/CO/CO2 ratios help in differentiating source signals?

AC **(L523-526)**: The ratios provide limited guidance on source sector revisions due to complexities, including errors from smaller footprints and highly variable emission factors. Since this study focuses on national total emissions, it highlights the national emission gaps rather than sector-specific gaps. In CHN, BC emissions, which require reduction, are revised by identifying the dominant sources based on magnitude and activity levels. For CO and $CO_2$, whose model gaps are harder to identify the source attributions, we consider that an approach involving updating the emission factors in sectors where BC needs revision. Additionally, results for the CHN case point to missing $CO_2$ sources beyond those shared with BC. This point was added to Sect. 3.2.2.

RC: Figure 6: Any explanation on why the model does not produce as much variability
(concentration range) as observations will be good to add.

AC **(L338-341, 425-428)**: The model exhibits much less variability than the observations in PHL, JPN (BC), and CHN (CO) cases, which could be due to several factors. The most evident reason is the coarse resolution as the model operates at ~0.5°, whereas the aircraft observations have a
much finer footprint (~0.01° per 15-second time step). Additionally, input emission inventories may lack detailed temporal variations, e.g. HTAPv2.2z in CHN case and REASv2.1 in PHL case, leading to differences in concentration ranges. Furthermore, atmospheric dispersion and mixing processes might be smoothed in the model, reducing sharp concentration contrasts observed in real measurements, especially when measurement locations are near emission hotspots or areas
with stronger variability, as in the PHL case. Errors in modeled meteorological factors also impact pollutant accumulation and dispersion, contributing to variability gaps in JPN, where air masses are mixed and not solely from JPN sources.
We included a detailed explanation for the PHL case (**L338-341**), as it shows the most significant difference between the model and observed concentration range, and in the caption of Figure 4
(**L425-428**).

RC: Figure 8: How did you calculate the uncertainty in red and green boxes?

AC **(L632)**: As discussed in Sect. 3.2.4 (Sect. 3.1.5 in the original manuscript), the uncertainties
of the emission estimates were propagated from (1) the uncertainties in the data of observationto-model ratios and emission ratios for multiple species, (2) the systematic errors of the instrument, (3) representation errors caused by the limited opportunities for aircraft observations to be made in terms of seasonal and spatial variabilities by comparing the spring and annual data, CEC region vs. the large footprint found at Fukue Island, and (4) an error for model-specific transport determined by comparing the BC concentrations simulated by the CMAQ model with the CAMx model system using the same emission data. These points have been included in the original manuscript (L580-588). We added a citation to **Sect. 3.2.4** in the caption of **Figure 8 (L632)**.

RC: Sec. 3.2: This section touches upon some of the points I raise above but a richer discussion in identifying the sources will be good to add. Consider using information from this section to inform inventory updates in the previous section. This does not need to be big changes but a short circling back would be useful.

AC **(L523-526, 613-615)**: As discussed above, while emission ratios are expected to reflect source characteristics and are used to diagnose national scale emissions, sector-specific information is often difficult to obtain. This is due to the complexity of regional anthropogenic activities, leading to greater errors from smaller footprints and highly variable emission factors. Therefore, we basically have to remain conservative. Nonetheless, several implications from this and our previous studies were drawn. Fire emissions, such as in the THL case, tend to exhibit a more uniform regional pattern, making them relatively easier to predict. A practical approach for inventory updates is to focus on dominant sources. In CHN, BC inventory can be improved by targeting major sources, specifically raw coal used in traditional cooking and heating stoves, based on their high emission factors and activity levels in the ECLIPSEv6b inventory (discussed in Sect. 3.2.2). Similarly, CO and $CO_2$ emission factors could be updated in the same sectors where BC requires revision, as well as other potentially missing sources.
We add a brief summary at the end of **Sect. 3.2.2** and **3.2.4**.

RC: Sec. 4: Consider adding recommendations on **how inventory-developers can directly utilize** such campaigns' information to tune up or down in this section.

AC **(L644-646)**: We added brief summary sentences to the Conclusion.

RC: L653: Add discussion on whether there is any chance or reason why the BC/CO ratios in the model and observations are similar due to completely different reasons, such as rapid BC deposition in the model offset by higher BC emissions?

AC **(L454-456)**: Though considered, it would be difficult to reproduce the observed BC/CO ratios in the model by changing deposition parameters in a reasonable range. The horizontal and vertical distributions of BC and CO at layers below the measurement altitudes (not shown) showed no indication of increased emissions followed by rapid deposition in the model. The similarity in the BC/CO ratio between the model and observations (e.g., the PHL case in E-AS-03), despite concentration gaps, likely results from appropriate BC and CO emission factors but insufficient activity levels in REASv2.1. We added some sentences for clarification in the caption of **Figure 6**.

---

## Author Comment (AC2)

Response to Referee Comments #2 on manuscript egusphere-2024-2064:

"Downward and upward revisions of Chinese emissions of black carbon and CO in bottom-up inventories are still required: an integrated analysis of WRF/CMAQ model and EMeRGe observations in East Asia in spring 2018"

5   Referee Comment (RC): **General remarks and questions**

This preprint reports aircraft measurements of black carbon (BC), CO, and $CO_2$ measured in the lower troposphere eastward along the Asian continent over the sea and around Japan and the Philippines. Pollution plumes with elevated levels of CO and BC are investigated and the observed concentrations are compared with predictions from the WRF-CMAQ model. Observed

10   to modeled concentration ratios derived from this are proposed as correction factors for emission inventories used in the model. Such an investigation of emission inventories is useful to achieve improved air quality predictions in the regions studied, and the aircraft measurements presented here provide a rare and good opportunity to extend analyses from ground based measurements.

Unfortunately, the presentation of the scientific approach and results in this paper is somewhat

15   confusing and difficult to understand in places. The following aspects are not addressed clearly enough for general understanding and require a more detailed discussion. Major revisions are required before the paper is suitable for publication in ACP.

Author Comment (AC): We appreciate the referee's thorough review and constructive feedback on our manuscript. We have addressed all concerns through substantial revisions, ensuring clearer

20   presentation of the scientific approach and results. The manuscript now provides enhanced clarity, improved methodological descriptions, and a more detailed discussion. Below, we provide detailed responses to each point raised and outline the corresponding revisions. Please note that the structure of the manuscript has been changed based on regions according to the comments of the first referee. Section 3 (Results) now includes 2 subsections: 3.1 for three polluted cases (THL,

25   PHL, JPN) including comparisons of concentrations and residual ratios; 3.2 for CHN case including comparisons of concentrations, residual ratios, and estimate for the emissions, with a separate part for uncertainty assessments. Section titles were also renamed. Revised points are highlighted in the manuscript. Minor textual adjustments and reconstructed text are not highlighted.

30   RC: (1) To what extent can CO, $CO_2$, and BC be considered as inert tracers that are essentially only subject to transport and deposition after emission on a 5d time scale, as considered in the

backward trajectory calculations. To what extent could other chemical or physical processes influence their concentrations observed on the aircraft?

AC (Revisions made in **Lines 171-179, 206-218**): Note that the time it takes for the air mass to travel from the sources to the observation points for the case studies is less than five days, typically < 40−120 hours from CHN, 0−24 hours from Manila (PHL), 10−120 hours from the broad fire region near THL, 0−36 hours from JPN (mentioned in Sect. 2.3., L196-198). During transport over less than 120 hours, CO, $CO_2$, and BC remain largely inert. CO undergoes slow oxidation with OH as its atmospheric lifetime is several weeks. As a stable gas, $CO_2$ is highly inert, particularly when transported at high altitudes without contact to vegetation, with deposition playing a minor role. BC amounts are affected by deposition. VOCs, including biogenic, undergo photochemical oxidation processes involving $NO_x$, leading to the formation of $O_3$ and secondary organic aerosols (SOA), which can accelerate BC aging. However, in polluted regions with strong emissions, this effect remains minor (**Sect. 2.2, L171-179**).

Since the flights were conducted away from rain events, the accumulated precipitation along the trajectories (APT), as derived from HYSPLIT rainfall data, helps to exclude data significantly affected by wet deposition. A detailed analysis of APT was conducted to assess the influence of rain on emission characteristics, applying APT criteria for the air masses from CHN and THL (APT 72 h < 1 mm), PHL E-AS-06 and E-AS-10 (APT 24 h = 0 mm), PHL E-AS-03 S2 (APT 10 h = 0 mm), JPN (APT 36 h = 0 mm) based on their travel time to the aircraft (CHN 40−120 h, THL 10−120 h, PHL E-AS-03 S2 0−10 h, E-AS-06 and E-AS-10 0−24 h, JPN 0−36 h; Figure S6, Table S6, S7). The APT 72 h < 1 mm criterion has been previously analyzed and showed no large influence of wet deposition on emission-concentration relationship (Kanaya et al., 2016, 2020), whereas the APT = 0 mm criterion aims to completely exclude the effect of rain. In addition to trajectory calculations, we carefully discuss deposition, dilution, and convection effects, as well as the representativeness of the air masses in the manuscript (**Sect. 2.3, L206-218**).

We provided additional discussions in **Sect. 2.2** and **2.3**.

RC: (2) Which source regions were exactly investigated using the approach with model calculations and flight measurements? Line 108 states: "Pollution plumes from major population centers in Asia were detected during parts of flights." Which major population centers are specifically meant? Asia is a very large continent.

AC (**L111-113**): We added the specification for Gulf of Thailand, Manila (Philippines), Osaka (Japan), and Central East China including Beijing, Hebei, Henan, Shanghai.

RC: Generalized statements such as "The results suggested that downward and upward revisions of Chinese emissions ... are required" (see abstract) is VERY general and should be formulated more cautiously and refer specifically to the regions or areas investigated.

AC (**L31**): The results suggest that revisions to Chinese emissions inventories are necessary, with a focus on Central East China during the polluted spring season, where our trajectory analysis indicates significant influence. Although our trajectories cover only Central East China for the aircraft observations, combining results from our previous study at Fukue Island with extended temporal and geographical coverage enabled discussion about the national-scale emission estimates. Associated uncertainties in extrapolating our findings to larger scales in terms of seasonal and spatial variations have been included. We added in the abstract where the inventory revision should focus on based on this and our previous studies.

RC: The authors should also consider making the title of the paper more specific.

AC: We made a change to the **title** i.e. "Assessing BC and CO Emissions from China Using EMeRGe Aircraft Observations and WRF/CMAQ Modeling"

RC: Also, to what extent could the findings be limited in time, e.g., only for the season or year investigated? How representative were the environmental conditions (fire activity, traffic, meteorology) prevailing during the flight campaign for the years before and after?

AC (**L456-464, 607-610**): Although the aircraft data for the China case was collected in spring, our estimation method was applied to all of China in 2018, accounting for representative uncertainties, including seasonal variations. Note that the uncertainty associated with representativeness has been discussed in the original text (**Sect. 3.2.4,** L582-585), additional explanation was added at **L609-610)**, supported by our previous year-round investigations (Kanaya et al., 2016, 2020). The environmental conditions during the campaign ensure the captured plumes were representative of dominant emission sources and seasonal patterns. Fire emissions account for only ~10% of anthropogenic emissions in China (7% for BC and 13% for CO, according to GFED and HTAPv2.2z inventories used in CMAQ), the detected pollution plumes by the aircraft predominantly originate from Chinese anthropogenic sources. Trajectory analysis indicates that the observed polluted plumes travelled from Chinese source regions starting around Thursday, 22 Mar 2018, and took 2–4 days to reach the aircraft. Consequently, the probed air masses were influenced by the typical weekday emissions from the transportation sector. The analyzed plume originating from a major population center in China was one of the most significant air pollution transport events recorded also by our long-term ground-based

observations at Fukue Island in western Japan over the past 10 years. The event was regarded as typical regarding its transportation route and negligible deposition loss (**Sect. 3.2.1, L456-464**). The annual and nation-wide emission estimates appear to be extrapolations using a limited set of aircraft observations. This extrapolation is supported by the results from our previous analysis, which used long-term ground-based observations (Kanaya et al., 2020) (**Sect. 3.2.4, L607-609**). This point has been clarified in the revised text. Our conclusion is obviously limited to the year of observations (2018).

RC: (3) The title of the paper states "Downward and upward revisions … are still required…". Is this statement still valid approximately seven years after the aircraft campaign? Have there been any developments in East Asia since then with regard to biomass burning, industrial emissions, or traffic that would warrant a new investigation?

AC (**L618-621, 622-626**): The estimated emissions and revision suggestion in this study apply to China in 2018, aligning with broader trends indicating a decline in BC emissions since 2010 (Kanaya et al., 2020) and still low levels in 2019–2020 (0.6 Tg yr$^{-1}$) according to the translation from TCR-2 CO data. However, updated inventories, such as CEDS v2021-02-05 and MEICv1.0 (Zhang et al., 2021), continue to show BC emissions exceeding 1 Tg yr$^{-1}$ from 2018 to 2020, reinforcing the need for a downward revision (**Sect. 3.2.4, L618-621**). The revised manuscript title "Assessing BC and CO Emission from China Using EMeRGe Aircraft Observations and WRF/CMAQ Modeling" does not overstate the results.

Changes since 2018, particularly in biomass burning, may influence emission trends. China's 13th Five-Year Plan for Biomass Energy Development (2016) promotes biomass energy, with further emphasis on biomass heating in the IEA Bioenergy Country Report (2021) and China Energy Transformation Outlook 2023 (COP28, December 2023). Meanwhile, vehicle exhaust remains a major pollution source despite advances in electrification and emissions control. These evolving factors could shift BC, CO, and $CO_2$ trends in uncertain directions, warranting future investigations. Nevertheless, the inventory revision suggestions for 2018 in this study remain worth considering (**Sect. 3.2.4, L622-626**).

We added a discussion in the last part of Sect. 3.2.4.

RC: (4) There is only very limited information given on the EMERGE campaign and the aircraft measurements. A brief overview should therefore be provided in Section 2.1. The following information would be of interest. Why was springtime chosen for the campaign?

AC (**L106-109**): We provided more detail for the EMeRGe campaign. Springtime in East Asia is characterized by peak emissions from biomass burning in Southeast Asia as well as by continental outflow from China that carries pollution to observable regions.

130 RC: What altitude range was covered during the measurement flights and what altitude range was specifically evaluated?

AC (**L114-115**): Flight altitudes during the measurements ranged from ground level to approximately 12 km, with ranges around 0.3–2 km were specifically evaluated for designated polluted areas.

135 RC: Which measuring instruments were used (manufacturer, model) and what are their total measurement errors (precision, accuracy)?

AC (**L117-118, 126-129**): We added more details on measuring instruments in Sect. 2.1.

RC: Which particle sizes were detected by the BC measuring device and how was the photometer signal converted into mass concentration? For which reference conditions (T, p) do the specified

140 mass concentrations apply?

AC (**L118-125**): SP2 detects BC cores with mass-equivalent diameters of 70–500 nm by measuring time-dependent scattering and incandescence signals as particles cross a Gaussian-shaped laser beam ($\lambda$=1064 nm) (Schwarz et al., 2006). Avalanche photo-diode detectors capture scattering and incandescence at high and low gain across two wavelength ranges (350–800 nm

145 and 630–880 nm). Particles scatter light based on optical size, while BC-containing particles absorb the laser, heating to ~4000°C and emitting incandescence. The peak intensity of this signal is linearly proportional to BC mass, calibrated using a correction factor (Laborde et al., 2013). BC concentrations were normalized to standard temperature and pressure (STP, $T_0$ =273.15 K, $p_0$ =1013.25 hPa). The SP2 incandescence signal was calibrated using size-selected fullerene soot

150 particles at the beginning, during, and end of the campaign (Holanda et al., 2020). We incorporated the explanation into the text and included two references (Schwarz et al., 2006 and Laborde et al., 2013).

RC: Were other compounds (e.g., NOx, VOCs) measured and used in this work, e.g., for the identification of plumes, apart from BC, CO, and CO2?

155    AC (**L130-131**): Other compounds, including $NO_x$, OVOC (HCHO) and VOCs, were measured during the campaign but not utilized in this study. While $NO_y$ was initially considered to investigate, its complex mechanisms during transport led to its exclusion.

RC: A PTR-MS instrument is mentioned in line 112. What was measured by this instrument and were the corresponding data used in any way?

160    AC: The PTR-MS is used to measure VOCs but these measurements not used in this study. We removed this sentence in the revised manuscript.

RC: (5) Section 2.2: which processes (apart from emission, transport, and deposition) play a role for the simulation of BC, CO, and $CO_2$ on the 5d time scale of the backward trajectories investigated? Table 1 lists various types of emissions. Apart from direct emissions of BC, CO,
165    and $CO_2$, do other emitted substances, for example biogenic VOCs mentioned in the table, play a role? If so, which chemical processes were involved? Which processes represented in the AERO 6 aerosol module are relevant for the modeled results of BC in this study? What role do the volcanic emissions listed in Table 1 potentially play?

AC (**L151-153, 171-178**): During the transport of about 5 d, CO, $CO_2$, and BC are effectively
170    inert. CO undergoes slow oxidation with OH, with a lifetime of weeks. VOCs (and biogenic VOCs) undergo photochemical oxidation processes involving $NO_x$, leading to the formation of $O_3$ and secondary organic aerosols (SOA), which in turn increases the BC aging, increasing hygroscopicity and enhancing removal through deposition. Although oxidation of biogenic VOCs can be an important source of CO, the plumes analyzed in this study originated from metropolitan
175    areas, suggesting only a minor contribution from biogenic sources. While $CO_2$ is absorbed by vegetation and marine phytoplankton, the $CO_2$ levels at altitudes of aircraft observations would be minimally influenced (**Sect. 2.2, L171-173, 175-178**).

In AERO6, BC concentration is unaffected by chemical or heterogeneous reactions, while coagulation alters particle size without affecting BC mass. Dry deposition loss of BC has been
180    tested as negligible during transport over 3 days (Fig. 6, Kanaya et al., 2016) (**Sect. 2.2, L151-153**). Wet deposition was found negligible in cases where the total amount of rain along the trajectory after the emission occurred is less than 1 mm (Kanaya et al., 2016, 2020). The analyzed measurements are in periods which are essentially free of wet deposition as indicated by the accumulated precipitation. Though the wet deposition scheme in the model may need to be tested
185    and revised, this dependency would have had minimal influence on the analysis and conclusions of this study (**Sect. 2.2, L173-175**).

The volcanic entry was **removed from the table**.

RC: (6) For each model component listed in Table 1, references should be provided where a description or corresponding data can be found.

AC: We added the references for each model component in **Table 1**.

RC: (7) In Section 2.4, the general description of the concept of "residual ratios" and "baseline values" is hard to understand and should be rephrased for clarity.

AC (**L223-225** and **261**): We adjusted the text for "residual ratios" (Sect. 2.4) and "baseline value" (Sect. 2.5) for clarification.

**Minor Comments**

RC: Abbreviations (EMERGE, WRF-CMAQ, GFED, HTAPv2.2z, etc.) should be defined at the beginning of the paper when they are first used.

AC: We have added the abbreviation definitions in the abstract and other relevant sections.

RC: Line 42: Typo "nanometre-sized"

AC (**L44**): It was changed to "nanometer-sized"

RC: Lines 96-97: The sentence is unclear. Which other pollutants (apart from CO, CO2, BC) were investigated?

AC (**L88**): We restructured the paragraph, this sentence became part of the study target (2) statement, and the old phrase was omitted.

RC: Line 105: Typo "Deutches"

AC (**L110**): It was modified to "Deutsches"

RC: What is the meaning of the given quantities (e.g., max = 1.3 mm) for APT3 in lines 340 and 575?

AC (**L197, 208-211, 484-485**): We changed the term to APT 72 h (accumulated precipitation along 72-hour trajectories). Previous studies have shown that a low APT 72 h value (e.g., < 1 mm) indicates weak wet deposition, which does not significantly influence emission-specific

characterization (Kanaya et al., 2016, 2020) (**Sect. 2.3, L210-211**). As the CHN air masses typically travel for 40–120 hours from source to the aircraft, APT 72 h was selected as the empirical criteria that the wet deposition becomes important for the CHN air masses (Kanaya et al., 2016, 2020) (**Sect. 2.3, L197**). We applied APT analysis using the criterion APT 72 h < 1 mm for CHN case, and omitted the specified APT maximum quantities as stated in the unrevised manuscript. Adjustments were made with the term expressions (**Sect. 3.2.2, L484-485**). Besides APT 72 h used for CHN air masses, we used different time range for other cases (**Sect. 2.3, L208-209**).

RC: Table S3 appears to have no use in the paper and should be deleted.

AC (**L507**): Table S3 describes in sufficient detail how the ECLIPSEv6b inventory assigns BC emission factors for each fuel type based on activity levels, abatement, and capacity control in CHN, supporting the discussion of dominant BC-emitting sectors in the original text (**Sect. 3.2.2**, L504-507). We have included the relevant citation for Table S3 (**L507**).

---

## Author Response (AR2)

**Authors' replies on Editor's Comments on ACP 2024-2064**

"Assessing BC and CO Emissions from China Using EMeRGe Aircraft Observations and WRF/CMAQ Modelling"

**5 **Public justification:**

Dear Phuc Ha,

The revised manuscript was re-evaluated by both reviewers who recommend publication in ACP. I agree that the presentation and description of the methods and results has improved and concerns of the referees have been well addressed. One minor change is still necessary before the manuscript can be published. The abstract of the manuscript is too long and does not comply with ACP's author guidelines (https://www.atmospheric-chemistry-and-physics.net/policies/guidelines\_for\_authors.html). Abstracts should have fewer than 250 words and provide a concise and accessible summary of the purpose, results, and implications of the research. Please revise the abstract accordingly.

15 Best regards,

10

Andreas Hofzumahaus

**AC:** We thank for your taking care of our manuscript. We carefully revised the abstract to meet the journal's criteria. Other changes including typos, co-authors' information change, figures updates, are also highlighted.

Lines 14 – 28: abstract revised

Line 8: co-author's affiliation updated

Line 302: typo

25 Lines 389 - 390: typo

Lines 534 – 560: changed according to changes in Figure 8

Line 432: Figure 6 was changed for better visualization, all the styles and legends were kept.

Legend entries for the ranges of sub-air masses were added.

Line 613: typo

Line 624 – 629: Figure 8 was changed for better visualization and more consistent with Figure 6, all the styles and legends were kept except marker styles for inventory lines changed from squares to circles. Legend entries for the uncertainty ranges were added. Caption was slightly changed.